# InFlux: A Benchmark for Self-Calibration of Dynamic Intrinsics of Video Cameras

**Erich Liang    Roma Bhattacharjee**\*  **Sreemanti Dey**\*  **Rafael Moschopoulos**
**Caitlin Wang    Michel Liao    Grace Tan    Andrew Wang**
**Karhan Kayan    Stamatis Alexandropoulos    Jia Deng**

Department of Computer Science, Princeton University
{erliang, rb4785, sd9968, gm3460, qw3971, ml4119,
gt8072, aw3948, kk2285, sa6924, jiadeng}@princeton.edu

## Abstract

Accurately tracking camera intrinsics is crucial for achieving 3D understanding from 2D video. However, most 3D algorithms assume that camera intrinsics stay constant throughout a video, which is often not true for many real-world in-the-wild videos. A major obstacle in this field is a lack of dynamic camera intrinsics benchmarks–existing benchmarks typically offer limited diversity in scene content and intrinsics variation, and none provide per-frame intrinsic changes for consecutive video frames. In this paper, we present Intrinsics in Flux (InFlux), a real-world benchmark that provides per-frame ground truth intrinsics annotations for videos with dynamic intrinsics. Compared to prior benchmarks, InFlux captures a wider range of intrinsic variations and scene diversity, featuring 143K+ annotated frames from 386 high-resolution indoor and outdoor videos with dynamic camera intrinsics. To ensure accurate per-frame intrinsics, we build a comprehensive lookup table of calibration experiments and extend the Kalibr toolbox to improve its accuracy and robustness. Using our benchmark, we evaluate existing baseline methods for predicting camera intrinsics and find that most struggle to achieve accurate predictions on videos with dynamic intrinsics. For the dataset, code, videos, and submission, please visit `https://influx.cs.princeton.edu/`.

## 1 Introduction

Camera intrinsics, which describe the projection of 3D points onto a 2D image plane, play an essential role in many 3D algorithms with important real-world applications. Intrinsics are essential for accurate depth estimation in robotics and for seamless virtual-real integration in AR/VR and CGI.

Despite the importance of accurate intrinsics, many 3D algorithms [27, 31, 32, 21, 20, 15] assume that intrinsics stay constant over all input images in a video, which fails to hold for many in-the-wild videos. For example, DSLR cameras with zoom lenses have changing intrinsics when adjusting the zoom or focus. Even smartphone cameras exhibit changing intrinsics due to autofocus. Hence, the assumption of constant intrinsics greatly limits the robustness and applicability of 3D algorithms.

The assumption of constant camera intrinsics stems from the lack of suitable evaluation data–currently, there exists no benchmark with dynamic intrinsic videos with per-frame ground truth, as collecting such data is highly non-trivial. Instead, most existing image benchmarks [9, 5, 29, 28] are collected under controlled conditions where camera intrinsics remain fixed throughout the recording. This simplification makes the data collection process significantly easier, because the camera only needs

---

\*Equal contribution.

39th Conference on Neural Information Processing Systems (NeurIPS 2025) Track on Datasets and Benchmarks.

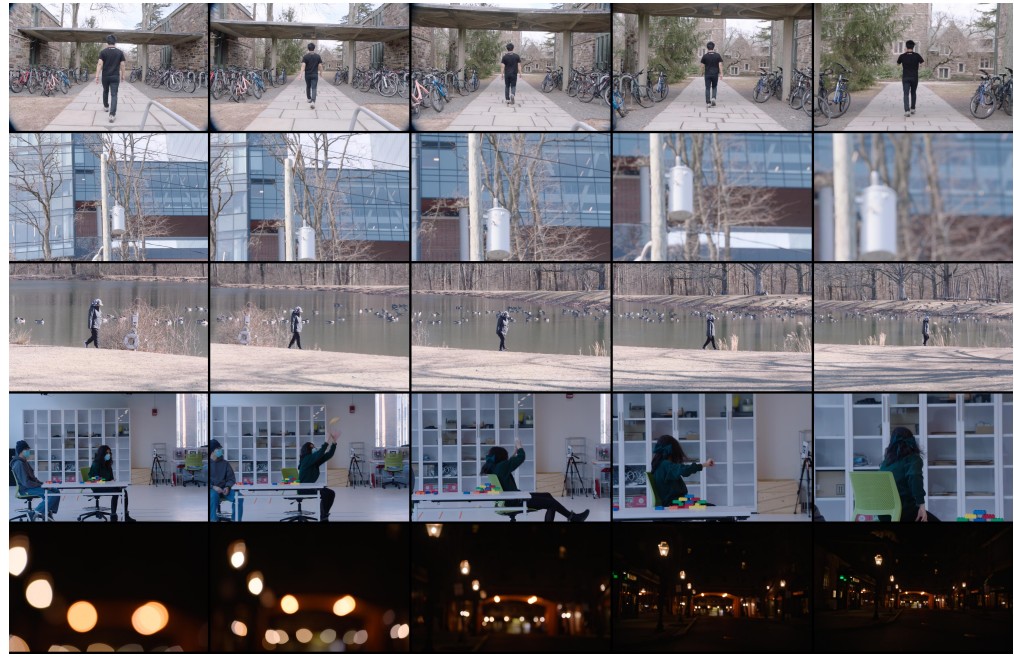

Figure 1: A gallery of InFlux, our real-world dynamic intrinsic video benchmark with per-frame ground truth intrinsics. It consists of of 143K+ frames across 386 videos and features highly diverse scenes, camera motion, and changes in intrinsics. We perform calibration experiments to construct per-lens lookup tables (LUTs) that map lens metadata to camera intrinsics. The LUTs are applied to our benchmark videos to generate ground truth per-frame camera intrinsics.

to be calibrated once, instead of on a per-frame basis. However, these benchmarks lack dynamic intrinsics often found in real-world videos, reinforcing the common assumption of constant intrinsics.

Among benchmarks that do include images with varying intrinsics, some suffer from scene diversity issues, while others provide inaccurate intrinsics. [16] provides intrinsics for videos filmed with around 40 types of wide-angle cameras, but these are raw calibration videos which are not representative of real-world footage. Another section of [16] compiles 300 internet images and reports the focal length of the lenses used per image. However, the deviation between lens focal length and the camera focal length used in intrinsics can be arbitrarily large, depending on the camera's focus settings.

In this work, we present Intrinsics in Flux (InFlux), a real-world benchmark of videos with dynamic intrinsics and per-frame ground truth annotations. We use zoom lenses that record per-frame lens metadata, which includes lens focal length (LFL) and focus distance (FD) which characterize a lens's optical state [1]. To get ground truth, we map each frame's LFL-FD values to intrinsics using a precomputed lookup table (LUT) for each lens. We construct these LUTs by calibrating each lens over many LFL-FD combinations and interpolating between the results. We populate the LUTs by collecting data using calibration boards of varying sizes and drone-based calibration. We extract intrinsics from the data using our in-house version of Kalibr [19], modified for better accuracy and robustness. These LUTs enable efficient per-frame retrieval of accurate intrinsics from lens metadata.

Using this approach, we obtain ground truth camera intrinsics for 143K+ image frames across 386 high-resolution real-world videos shot with varying camera intrinsics. Our video benchmark has high scene diversity, featuring 126 indoor scenes and 260 outdoor locations such as hallways, offices, natural landscapes, and urban environments. The videos feature diverse camera motions and intrinsic adjustments induced by changing the camera's zoom, focus, or both. We also include natural cinematic shots, such as smooth panning with zoom and dolly effects, which are commonly found in documentaries and films. A gallery of our benchmark images can be found in Fig. 1.

Using InFlux as evaluation data, we run several baseline methods [27, 10, 34, 14, 23, 37] that predict camera intrinsics given input images. Evaluation results indicate that current methods struggle to accurately predict per-frame camera intrinsics on our benchmark. In summary, our contributions are:

- We introduce the first benchmark that provides per-frame ground truth camera intrinsics for real-world videos with dynamic intrinsics.
- We provide a diverse set of 386 high-resolution videos covering over 143K+ annotated frames. These videos capture diverse indoor and outdoor environments and exhibit a wide range of intrinsic changes and camera movements.
- We extend Kalibr to significantly improve its accuracy and robustness on our data.
- We evaluate baseline methods on our benchmark and show that they struggle with dynamic intrinsics prediction on our benchmark.

## 2    Preliminaries

**Camera Focal Length (CFL)** is the distance between a camera's optical center and its imaging plane, measured along the optical axis. When measured in units of pixels, CFL is equivalent to the $f_x$ and $f_y$ terms in the camera intrinsics matrix.

**Lens Focal Length (LFL)** is defined as the CFL of a camera when the camera is focused at infinity [25]. Lenses are typically described by their LFL values (e.g., "50mm lens"). Usually, CFL is not equal to the LFL because the focus of a camera is not at infinity in most real-world images.

**Lens to Object Distance (LTO)** is the distance between a camera's optical center and the object in focus, measured along the optical axis. This is equivalent to depth in computer vision.

**Focus Distance (FD)** is the distance between a camera's sensor plane and the object in focus, measured along the optical axis. It is equal to the sum of CFL and LTO.

**Zoom Lenses** can adjust a camera system's intrinsics during filming via specialized rings that move internal lens groups. The zoom ring sets the LFL, while the focus ring sets the FD, with markings on the camera barrel indicating their values. In lenses that report metadata, the recorded focal length and focus distance correspond to LFL and FD.

## 3    Related Works

### 3.1    Camera Intrinsics Benchmarks and Datasets

**Real-World Intrinsics Benchmarks** are included in many popular vision benchmarks [9, 5, 29, 28], which provide camera intrinsics for their images and videos. But unlike our benchmark, these benchmarks lack changing camera intrinsics since they use a fixed lens throughout data collection. One component of [16] captures videos from around 40 types of wide-angle cameras. However, these videos are only limited to calibration videos, which lack realism and scene diversity compared to our benchmark. Moreover, their intrinsics remain constant within each video and only vary across different recordings. [16] also scrapes 300 high-resolution internet images and reports the lens focal length for each image. However, this is inaccurate, as lens focal length can differ significantly from the actual camera focal length used in camera intrinsics, depending on the camera's focus settings. In contrast, our benchmark provides accurate camera intrinsics by constructing LUTs that take into account camera zoom and focus settings.

**Synthetic Intrinsics Datasets** such as [35, 16, 24] could in theory support training for dynamic intrinsics prediction, but are unsuitable as benchmarks due to the visual sim2real gap. But even as training data, current options fall short: [35] uses the same intrinsics for all images, [16] provides pairs of distorted and rectified images but does not provide ground truth intrinsics, and [24] lacks scene diversity because it only generates calibration board sequences.

### 3.2    Camera Calibration Techniques

**Calibration Board Methods** such as [18, 33, 11, 19, 13, 2] produce camera intrinsics using images of planar calibration targets with known 3D geometry. Among these methods, Kalibr [19] is recognized for its superior accuracy and consistency [12]. But in practice, it struggles with convergence due to

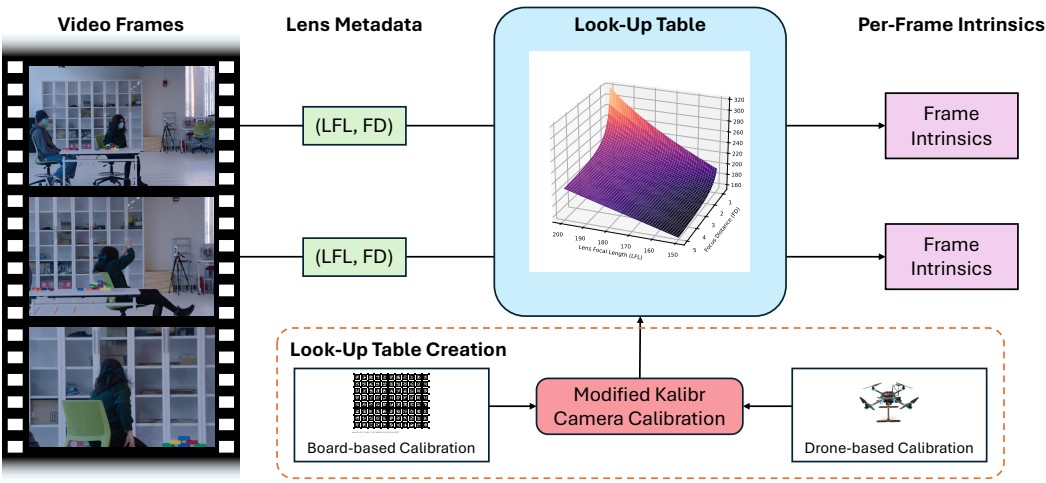

Figure 2: A visual representation of our data collection process for our real-world benchmark of videos with dynamic camera intrinsics. For each video frame, we record its LFL and FD values. These are used to query the LUT for the lens used to obtain corresponding ground truth intrinsics. To construct each lens's LUT, we perform a set of board-based and drone-based calibration experiments and apply an interpolation strategy to allow for querying of novel LFL-FD pairs.

focal length initialization issues and the principal point drifting significantly away from ground truth. We modify Kalibr to address these issues to produce accurate ground truth camera intrinsics.

**Large Field of View (FOV) Calibration Methods** use other calibration targets instead of board-based ones, because manufacturing sufficiently large and flat calibration boards is impractical. Like us, [36] uses a drone with Real-Time Kinematic (RTK) positioning as a moving calibration target. By aligning the drone's 2D image coordinates with its corresponding 3D RTK positions, camera intrinsics can be optimized via reprojection loss. However, [36] only models an ideal pinhole camera, omitting distortion commonly present in real-world cameras. In contrast, we optimize for full intrinsics using the Brown–Conrady distortion model [3].

**Intrinsics Estimation Techniques** estimate camera intrinsics directly from in-the-wild images or videos without the use of calibration targets. COLMAP [27, 26, 7, 17] and DroidCalib [10] predict camera intrinsics for input video by utilizing correspondences between nearby frames. However, both assume that intrinsics stay constant throughout the entire video. GeoCalib [34], Perspective Fields [14], UniDepthV2 [23], and WildCamera [37] predict intrinsics for individual input frames, but applying this to each frame in a video can produce a sequence of intrinsics that do not change smoothly or interpolate well.

## 4 Methodology

Our real-world benchmark aims to record videos with dynamic intrinsics while logging ground truth intrinsics for every frame. One naive approach would be to perform a full camera calibration for every frame. However, this would be computationally expensive and time consuming. Moreover, recalibrating between frames would require recording the video in stop-motion style, disrupting natural motion and undermining the realism of the video. This poses a key challenge: how can we obtain accurate per-frame ground truth intrinsics without disrupting video continuity?

To address this challenge, we use special cameras and lenses (§Sec. 4.1) that record per-frame lens metadata. This includes LFL and FD, which parameterize a lens's optical state [1] and uniquely determine its intrinsics. Hence, if we precompute a LUT for each lens that maps pairs of LFL-FD values to intrinsics, we can retrieve accurate per-frame intrinsics without disrupting video continuity. This shifts the burden of calibration from a per-frame operation to a one-time LUT construction step.

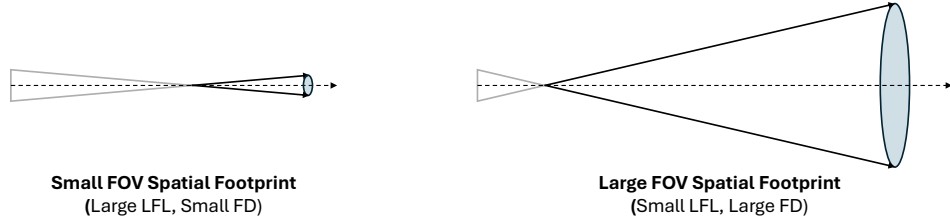

**Small FOV Spatial Footprint**
**(**Large LFL, Small FD)

**Large FOV Spatial Footprint**
**(**Small LFL, Large FD)

Figure 3: Examples of how FOV spatial footprint (FSF) varies in size depending on LFL-FD settings. For full FOV coverage and accurate 2D detections, calibration targets must scale with FSF size. We use targets of different sizes to match the range of FSF size across different LFL-FD settings.

To create the LUTs, we perform calibration experiments for each lens across a wide range of LFL and FD values using calibration boards and drones (§Sec. 4.2). We process collected data with our in-house version of Kalibr [19], modified for improved robustness and versatility (§Sec. 4.3). The calibration results are stored in each lens's LUT, and we define an interpolation scheme over the results (§Sec. 4.4). This enables querying of per-frame ground truth intrinsics for our benchmark, even for LFL-FD settings not seen during calibration. See Fig. 2 for an overview of our method.

## 4.1 Camera Hardware

We use the ARRI Alexa Mini camera and two zoom lenses: Canon CINE-SERVO 17-120 mm PL Mount (canon17) and Fujinon Premista 80-250 mm (premista80). These lenses record /i Technology lens metadata, which contains per-frame LFL and FD values that uniquely determine camera intrinsics.

## 4.2 Calibration Experiments

To construct each lens's LUT, we cannot calibrate at every possible LFL-FD combination, as both are continuous variables. Instead, we select a subset of experiments to calibrate at using the criteria: (1) denser sampling at low LFL values, where 3D unprojection is sensitive to changes in intrinsics; (2) denser sampling at low FD values, where CFL deviates more from LFL; and (3) maximal LUT coverage. To achieve these, we sample LFL with exponentially increasing step size and FD roughly uniformly in inverse depth. See our supplement for full sampling heuristic details.

Camera calibration relies on 2D-3D point correspondences. As a result, for each experiment, the calibration target must meet several key requirements to ensure accurate results. First, the target must have known 3D structure–this is often a planar calibration board, which has easily described 3D structure. Second, the camera must accurately obtain 2D point detections. This implies that the target must be placed at the FD to stay sharp, and the target should be as large as possible to help with 2D detection precision. Third, the target must be moved around to cover the entire camera FOV, as this is especially important for accurately modeling distortion, which is more apparent at fringes.

These requirements imply that: (1) the FOV spatial footprint (FSF), defined as the 3D region within camera FOV around the selected FD, can vary greatly in size for different LFL-FD settings (see Fig. 3); and (2) the target must scale with FSF size to ensure accurate 2D detections. For small and medium FSF settings, we use calibration boards of various sizes. However, for large FSF settings, manufacturing a suitably sized board becomes impractical. In these cases, we instead use a more maneuverable drone calibration target. See Fig. 4 for the target types used per experiment.

### 4.2.1 Small to Medium FSF: Board-based Calibration

For these LFL-FD settings, we choose an appropriately sized AprilGrid board as the calibration target. Each of our four boards consists of an $8 \times 11$ array of AprilTags [22], and tag size scales proportionally to board dimensions. The board dimensions we use are $100 \times 75$ mm, $200 \times 150$ mm, $400 \times 300$ mm, and $800 \times 600$ mm. For each experiment, we use the largest board that fits entirely within the camera's FOV at the given FD. See our supplement for additional board details and visuals.

During calibration, we orient the board to excite all axes of rotation. For each orientation, we move the board through the FSF to cover the camera's FOV. We record calibration videos, so to aid keyframe

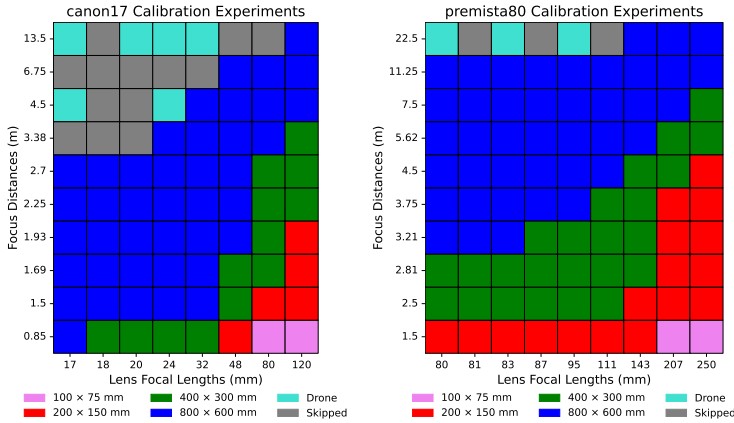

Figure 4: A visualization of the different calibration experiments performed to fill in the LUT for canon17 and premista80. Because different LFL and FD combinations yield different FSF sizes, we use a variety of different calibration targets that scale with FSF size.

extraction, we move the board jerkily between poses. This makes desired key frames sharp with many 2D detections, and makes in-between frames blurry with few detections. We use ANMS [4] based on 2D detection count and frame index to select keyframes. See our supplement for details.

As the FSF of LFL-FD combinations grows, board calibration becomes impractical. First, manufacturing and moving very large planar boards can be too expensive and cumbersome. Assuming a fixed camera height, the FSF can also grow until it extends into the ground, making parts of the camera FOV unreachable at the correct FD distance. When this occurs empirically in our experiments, we classify the LFL-FD combination as having a large FSF and use an alternative calibration target.

#### 4.2.2 Large FSF: Drone-based Calibration

To overcome the physical limitations of board-based calibration in large FSF settings, we use drone-based calibration. Drones are maneuverable in open sky, which helps avoid the FSF clipping issues found in board-based calibration. However, unlike calibration boards, drones are not inherently equipped for establishing 2D-3D point correspondences, which is necessary for camera calibration.

We enable 2D-3D point correspondence detection by equipping a Holybro X500 V2 drone with additional hardware. For accurate 2D detections, we mount a 3-watt red LED to the underside of the drone, which is salient and easy to localize when filming at night. For accurate 3D detections, we mount a Septentrio Mosaic X5 Real-Time Kinematics (RTK) chip to the top of the drone. This provides low-latency 3D position data with centimeter-level accuracy. To ensure sharp 2D detections and reliable 3D readings, we only record data when the drone is hovering still. We achieve this using a Raspberry Pi (RPi) onboard the drone that briefly flashes the LED on and logs RTK data upon receiving a remote SSH signal. See our supplement for more details.

To perform drone calibration for a given LFL-FD setting, we compute a 3D flight path that stays within the FSF and maximally covers the camera's FOV. We begin by using the thin lens equation to approximate the CFL. Based on this CFL and factors like camera height and orientation, we unproject the camera geometry to estimate the FSF. Since real cameras deviate from the thin lens model, we refine this estimate by flying the drone along a test path near the FSF boundaries and empirically scaling the pattern to fit the actual FOV. After FSF tuning, we generate a final flight path consisting of 24 hover points and upload it to QGroundControl for autonomous flight execution. During flight, we record video and monitor QGroundControl's UI to identify hover moments, at which point we send a signal to the onboard RPi to trigger data capture. See our supplement for details.

To process the collected data, we run a hue-based detection algorithm to identify video frames where the red LED is on. For these frames, we localize 2D drone location by fitting an elliptic contour to the LED's pixels and reporting the center [30]. When the LED is detected across consecutive frames, we only retain the first frame's 2D keypoint, as it is temporally closest to the corresponding RTK

reading. For RTK data, we convert from geodetic to Cartesian coordinates. Finally, we match our 2D keypoints to 3D RTK positions chronologically. See our supplement for details.

### 4.3 Calibration Software: Kalibr Modifications

To process calibration data, we use Kalibr [19] to compute camera intrinsics that populate our LUTs. Kalibr takes in 2D images of planar targets and outputs intrinsics following the Brown-Conrady distortion model [3]. Calibration occurs in four stages: (1) camera pose initialization via Perspective-n-Point; (2) CFL initialization using vanishing points (VP) [8]; (3) distortion initialization via Levenberg–Marquardt (LM); and (4) joint optimization of poses and intrinsics using LM. In the final stage, Kalibr processes each frame sequentially in a random order and performs outlier rejection. However, we observe issues with Kalibr's robustness and accuracy, which our modifications address.

Because LM is sensitive to initialization, Kalibr sometimes fails to converge when the VP-based CFL estimate is poor. We address this by replacing Kalibr's CFL estimate with a thin lens-based approximation, leveraging our privileged knowledge of the lens's LFL and FD.

During distortion initialization, the estimated intrinsics can also become unstable, potentially leading to poor initialization for the final LM optimization stage and eventual inaccurate results. This instability is often characterized by the principal point drifting too far from the image center, which is implausible given the quality of our camera hardware. To address this, we propose a "fixed point" initialization scheme, where we periodically reset the principal point back to the image center during iterations of LM for distortion initialization. See our supplement for details.

During the final stage of optimization, Kalibr processes input images sequentially in a random order to perform outlier rejection. Empirically, we find that this stochastic ordering occasionally produces intrinsics that deviate significantly from that of most runs. To reduce variance, we perform multiple rollouts of Kalibr and select a representative trial with the median aggregate result across several intrinsic parameters. Since this final joint optimization step uses LM, it can be sensitive to noise and produce inaccurate results. If the final result's principal point drifts too far from the image center, we revert to the intrinsics produced before the final LM step. See our supplement for details.

### 4.4 LUT Interpolation Scheme

Board-based experiments in our LUTs form approximately regular grids in LFL-FD space, with minor jitter in FD values. For grid regions enclosed by four board experiments, we apply trapezoidal bilinear interpolation over all intrinsics to assign values at intermediate LFL-FD points.

Drone-based experiments are more irregularly distributed. For non-grid regions, we perform Delaunay triangulation [6] using board and drone experiments as vertices. Within each triangle, we apply barycentric interpolation to assign intrinsics at intermediate LFL-FD points.

We intentionally avoid more sophisticated interpolation schemes, as they may introduce bias when modeling complex real-life lens systems. See Fig. 5 for an illustration of our interpolation. We also perform leave-one-out cross-validation experiments that show that our LUT interpolation is accurate and produces reliable ground truth. See our supplement for more details and experiment results.

## 5  InFlux Benchmark Diversity

Our real-world benchmark contains over 143K frames from 386 high-resolution videos with changing camera intrinsics. Using our LUTs, we can obtain per-frame ground truth intrinsics for virtually any video we can physically record, enabling high diversity in scene content, camera motion, and intrinsics variation. Of the 386 videos, 126 are shot indoors, covering scenes such as offices, hallways, and classrooms. The other 260 videos are filmed outdoors across various lighting conditions and times of day, featuring environments like natural landscapes, parks, and urban areas. The scenes feature a broad range of subjects–from static structures like buildings and furniture to dynamic elements such as people, vehicles, and animals–further enriching the dataset's diversity.

Camera motion ranges from static shots to controlled pans and tilts, fast sweeps, tracking and dolly movements, object scans, bumpy handheld footage, wild random swings, and other complex trajectories. We vary camera intrinsics in many ways, including fixed settings, controlled monotonic

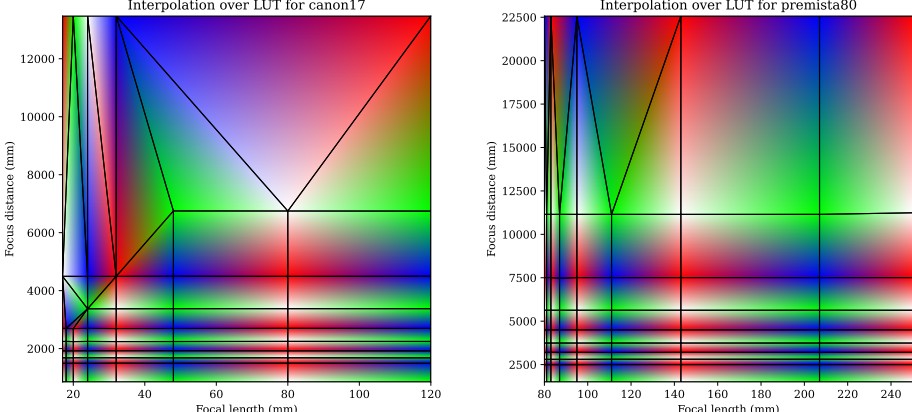

Figure 5: A visual representation of our LUT interpolation scheme, where the color of a point illustrates the relative effect of each vertex of the region it is in. In quadrilateral regions, bilinear interpolation is used, while in triangular regions, barycentric interpolation is used.

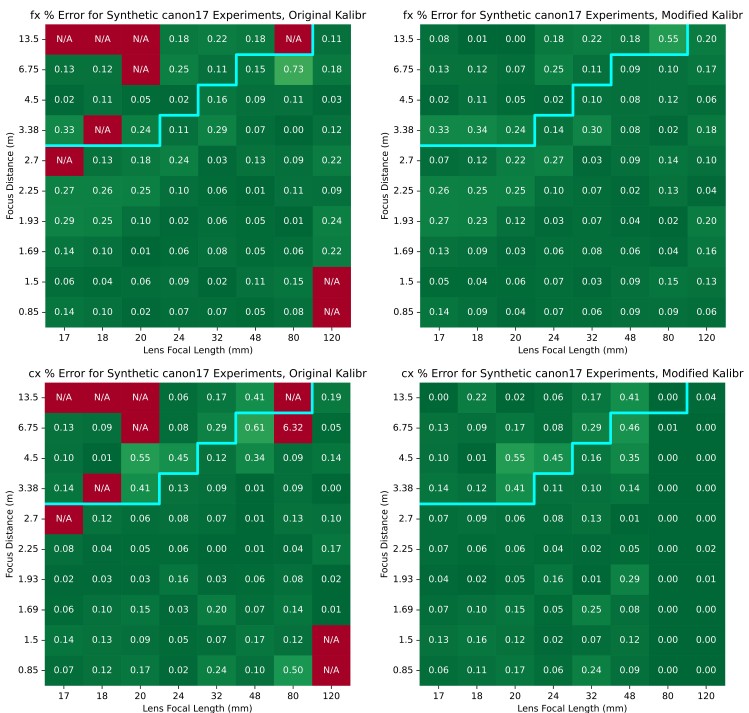

Figure 6: Synthetic canon17 experiment results comparing original Kalibr (left) and modified Kalibr (right). Our modifications help with convergence issues, improve accuracy, and reduce variance of predictions. Synthetic drone experiments are above the teal line; synthetic board are below.

changes in LFL and FD, periodic variations, and wild non-monotonic fluctuations. These changes may affect one parameter at a time, or both simultaneously in uncorrelated ways. We also include more structured intrinsics changes such as zoom-ins and dolly shots, as well as realistic, scene-driven adjustments like dynamically changing zoom to shift focus between subjects of interest.

Table 1: Performance of baselines for intrinsics prediction on our benchmark. We present mean percent errors for CFL and principal point parameters, and percent of frame-point pairs below a threshold EPE of 300 pixels. GeoCalib performs best but still has high EPE error.

| Method | % fx Error | % fy Error | % cx Error | % cy Error | % EPE < 300 px ↑ |
|---|---|---|---|---|---|
| GeoCalib [34] | 56.5 | 56.5 | 9.88e-2 | 2.04e-1 | 52.9 |
| WildCamera [37] | 45.6 | 46.9 | 5.04 | 6.39 | 47.2 |
| UniDepthV2 [23] | 50.6 | 51.1 | 1.61 | 2.58 | 46.1 |
| DroidCalib [10] | 68.1 | 70.0 | 10.1 | 15.7 | 28.0 |
| Perspective Fields [14] | 64.6 | 64.6 | 18.6 | 19.7 | 17.8 |
| COLMAP [27] | 1270 | 1280 | 1.12e-1 | 2.99e-1 | 7.85 |

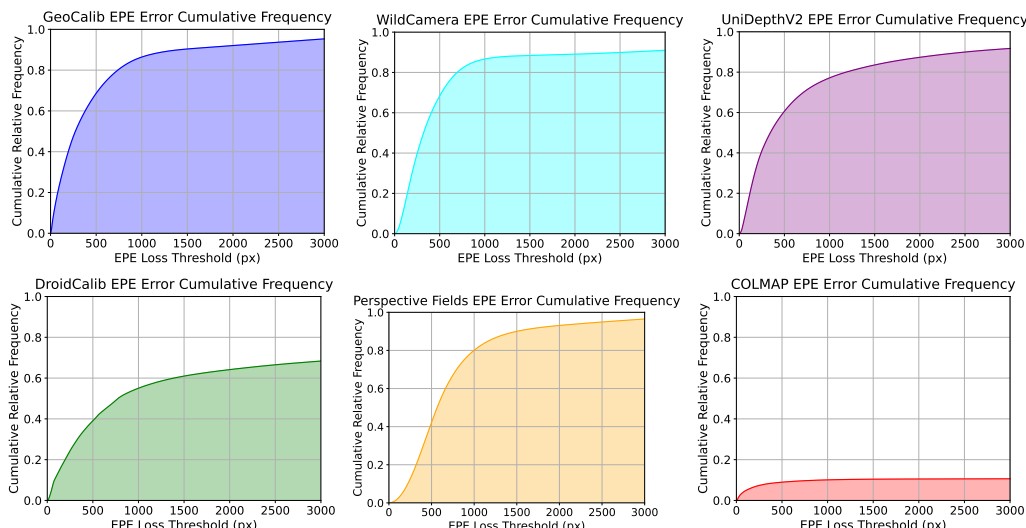

Figure 7: Plots of the cumulative relative frequency of EPE errors for each baseline method. GeoCalib, which predicts intrinsics frame by frame, has the best performance, but still has high EPE error.

# 6 Experiment Results

## 6.1 Evaluating Modified Kalibr: Synthetic Calibration Experiments

To evaluate the effectiveness of our modified Kalibr algorithm, we compare it to original Kalibr on a set of synthetic calibration experiments. We use Blender to render synthetic board and drone targets at LFL, FD, and distortion settings similar to those found during LUT creation. In Fig. 6, we report the percent errors of CFL and principal point predictions for both methods. Our modified Kalibr achieves consistently low error, in contrast to the original version, which exhibits occasional large error spikes. Moreover, our version converges reliably across all experiments, whereas the original Kalibr fails to do so. See our supplement for more details on experiment design and additional evaluation.

## 6.2 Baseline Method Results

Using InFlux, we evaluate six intrinsics prediction baselines: COLMAP [27], DroidCalib [10], GeoCalib [34], Perspective Fields [14], UniDepthV2 [23], and WildCamera [37]. We run COLMAP via [26]'s repository setup, using `PER_IMAGE` camera mode, `superpoint_max` [7] features, `lightglue` [17] feature matching, a value of 0.05 for `min_focal_length_ratio`, a value of 20.0 for `max_focal_length_ratio`, and a stride of 5. We run DroidCalib with default settings and apply its single intrinsics prediction to all frames. We run Perspective Fields, UniDepthV2, and WildCamera using `Paramnet-360Cities-edina-uncentered`, `unidepth-v2-vitl14`, and `wild_camera_all.pth` weights, respectively.

We perform evaluation in two ways. First, we compute the mean percent error of CFL and principal point between our ground truth and each baseline's predictions. Second, we sample 100K 3D points from [28]. For each frame in InFlux, we select the points within FOV and project them onto the camera sensor using both ground truth and predicted intrinsics. We report statistics on the end point error (EPE) between the two projections for each frame-point pair. See our supplement for details.

We find that COLMAP and DroidCalib fail to produce intrinsics predictions for some input frames. We run COLMAP with one camera per frame, but COLMAP still fails to produce intrinsics predictions for 92% of input frames. For frames it produces predictions for, COLMAP's CFL predictions are wildly inaccurate, resulting in high EPE error. DroidCalib fails to predict intrinsics for 15% of input frames. It often fails on videos with little motion, as it relies on presence of large amounts of optical flow for keyframe selection. GeoCalib achieves the best performance among the baselines. However, only 54.1% of its frame-point pairs achieve EPE under 300 pixels, whereas image dimensions are $3424 \times 2202$. See Tab. 1 and Fig. 7 for results, and our supplement for more details.

## 7 Conclusion

We introduce InFlux, a diverse real-world benchmark featuring per-frame ground truth intrinsics for videos with changing camera intrinsics. By mapping lens metadata to intrinsics via LUTs, InFlux enables the extraction of accurate per-frame intrinsics from virtually any recorded video. This allows our benchmark to capture a wide range of environments, camera motions, and intrinsics variations. Although camera intrinsics are essential for understanding 3D structure, we find that current baselines methods struggle to predict accurate intrinsics on InFlux. By releasing InFlux, we aim to advance the study of dynamic intrinsics in video and pave the way towards 3D methods that are robust to in-the-wild videos with dynamic camera intrinsics.

## 8 Acknowledgments

This work was partially supported by the National Science Foundation. Erich Liang was supported in part by the NSF Graduate Research Fellowship Program under Grant No. 2146752. We thank our friends and colleagues at Princeton University for their help with filming the benchmark.

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

# Appendix

## A    Additional Details on LFL and FD Sampling

### A.1    Experiment Sampling Strategy

As mentioned in our main paper, we select which calibration experiments to perform based on the following criteria: (1) denser sampling at low lens focal length (LFL) values, where 3D unprojection is sensitive to changes in intrinsics; (2) denser sampling at low focus distance (FD) values, where camera focal length (CFL) deviates more from LFL; and (3) maximal lookup table (LUT) coverage. The third criterion is self-explanatory. Hence, we will now explain the motivation for the first two criteria and why they encourage more sampling at lower LFL and FD values.

The first criterion encourages sampling at lower LFL regions in the LUT where 3D unprojection is sensitive to perturbations in intrinsics, as 3D unprojection is a key problem in many downstream 3D algorithms such as simultaneously localization and mapping (SLAM) and 3D reconstruction. Consider a fixed pixel and a fixed depth to unproject it to. When the CFL of the system is low, perturbing the CFL by a small amount will cause a large change in the 3D position the pixel is unprojected to. On the other hand, when the CFL of the system is high, a similar magnitude of CFL perturbation will cause a much smaller change in unprojection position. See Fig. A for a visualization. Hence, lower CFL corresponds to higher 3D unprojection sensitivity, and since lower CFL is generally correlated with lower LFL, we choose to sample more when LFL is low.

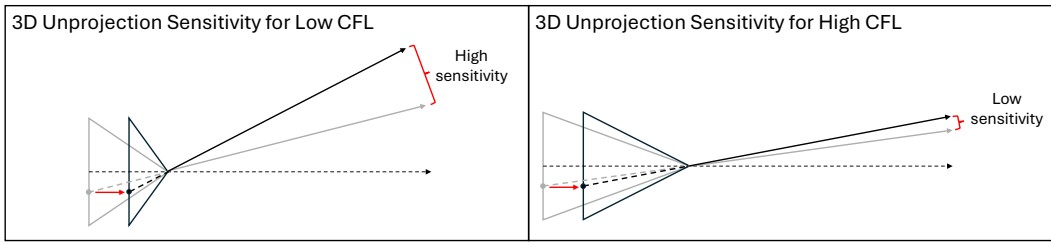

Figure A: Visualization of 3D unprojection sensitivity for different CFL values. When CFL is low, the 3D position a pixel is unprojected to has higher sensitivity to perturbations of CFL. Lower CFL is correlated with lower LFL, so we sample more when LFL is low.

The second criterion, which encourages sampling at lower FD values, is designed to capture more raw LUT data in regions where the CFL deviates more from the LFL. Recall that LFL is defined as the value of CFL when the camera system is focused at infinity. Hence, CFL deviates from LFL when the camera system's focus is closer than infinity, and for a thin lens model, this deviation grows as FD decreases. A real lens may not behave exactly the same as a thin lens, but a similar trend holds empirically. See Fig. B for a visualization of the thin lens case.

### A.2    Sampled LFL and FD Values

For each lens, we select a list of LFL and FD values. We then consider the set of experiments at all combinations of the chosen LFL and FD values.

**LFL Values.**  Each lens has a minimum and maximum LFL value, which we denote as $LFL_{min}$ and $LFL_{max}$, in millimeters. Based on these lens properties, we sample $k$ LFL values with exponentially increasing step size as follows:

$$\text{Sample lowest LFL value:}\quad LFL_0 = LFL_{min}$$
$$\text{Sample with increasing step size:}\quad LFL_i = LFL_{i-1} + 2^{i-1}, \quad \forall i \text{ s.t. } LFL_i < LFL_{max}$$
$$\text{Sample highest LFL size:}\quad LFL_{k-1} = LFL_{max}$$

Thus, for the canon17 lens, we select the following LFLs: 17 mm, 18 mm, 20 mm, 24 mm, 32 mm, 48 mm, 80 mm, and 120 mm. For the premista80 lens, we select the following LFLs: 80 mm, 81

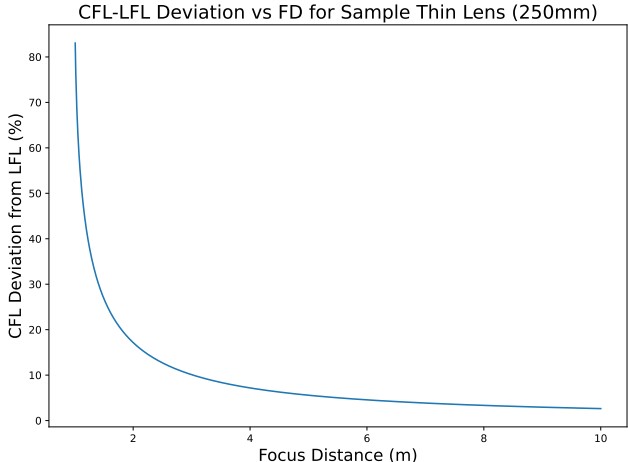

Figure B: Visualization of percent deviation of CFL from LFL for a thin lens with LFL of 250 mm, as a function of FD. In general, CFL deviates from LFL more when FD is low.

mm, 83 mm, 87 mm, 95 mm, 111 mm, 143 mm, 207 mm, and 250 mm.

**FD Values.** Each lens has a minimum FD value which we denote as $FD_{min}$, in meters, and has a maximum FD value of infinity. We also pick a lower-end FD value for each lens, which we denote as $FD_{lower}$. To sample more densely at lower FD while achieving coverage, we start out by sampling 9 values uniformly in disparity, with $\frac{1}{FD_{lower}}$ and $\frac{1}{\infty}$ serving as bounds. We then also pick the true minimum FD value. The full algorithm is as follows:

$$FD_0 = FD_{min}$$
$$\text{Set } D_i = \frac{10 - i}{9 \cdot FD_{lower}} \text{ for } i = 1, \ldots, 9$$
$$FD_i = \frac{1}{D_i} \text{ for } i = 1, \ldots, 9$$

For the canon17 lens, we have that $FD_{min} = 0.853$ m, and we pick $FD_{lower} = 1.5$ m, which yields the following FDs: 0.85 m, 1.69 m, 1.93 m, 2.25 m, 2.7 m, 3.38 m, 4.5 m, 6.75 m, and 13.5 m. For the premista80 lens, we have that $FD_{min} = 1.5$ m, and we pick $FD_{lower} = 2.5$ m, which yields the following FDs: 1.5 m, 2.5 m, 2.81 m, 3.21 m, 3.75 m, 4.5 m, 5.62 m, 7.5 m, 11.25 m, and 22.5 m.

## B    Additional Details for Board-Based Experiments

### B.1    Board Pattern and Manufacturing

As mentioned in our main paper, the board sizes we use are $100 \times 75$ mm, $200 \times 150$ mm, $400 \times 300$ mm, and $800 \times 600$ mm. The tag size for the smallest board size is 6 mm, with tag spacing of 1.8 mm, and these values scale up proportionally with board dimensions. The points detected are the corners of the main tags. Because we use a $8 \times 11$ pattern and each tag has 4 corners, there is a total of 352 points that can be detected in any frame. Our boards are manufactured by Calib.io. See Fig. C for visualizations of the different board sizes.

### B.2    Board Movement

During board calibration, we follow the recommendations of [19] by exciting all axes of rotation. We exclude rotation about the axis normal to the board surface, as it results only in in-plane image rotation. To ensure the board is rotated about the other two axes, we use a set of 5 board orientations to hold it at. The first orientation is with the board held parallel to the camera sensor, and the other four involve tilting the board at $\pm 45°$ about each of the other two axes. See Fig. D for a visualization.

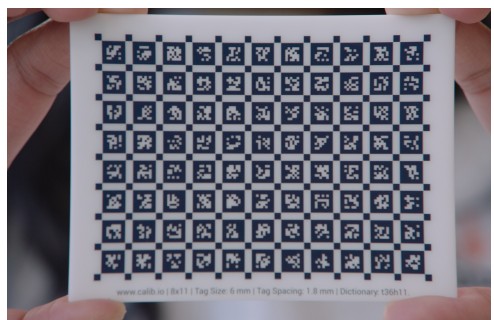
100 x 75 mm, 6 mm tags

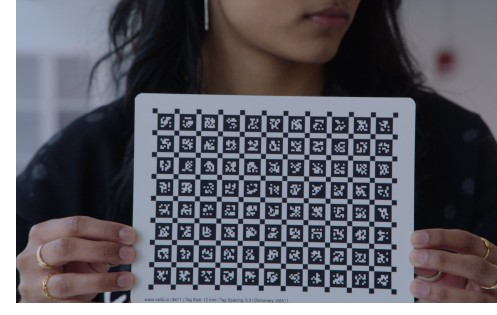
200 x 150 mm, 12 mm tags

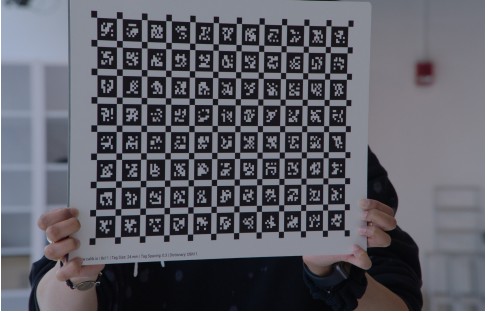
400 x 300 mm, 24 mm tags

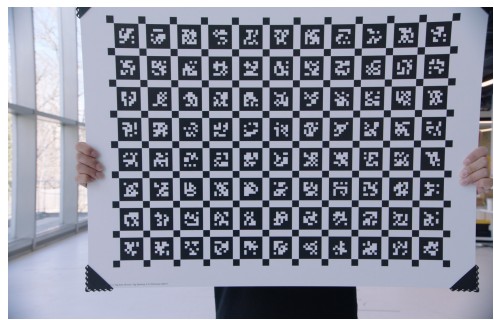
800 x 600 mm, 48 mm tags

Figure C: A gallery of the board sizes used for our board calibration experiments. Each board has the same tag layout that scales with board dimension. All four boards are manufactured via Calib.io.

For each board orientation, we move the board throughout the camera's field of view spatial footprint (FSF)–namely, the 3D region the camera's field of view (FOV) covers at the selected FD. While the number of intended key poses per orientation varies empirically, it typically ranges from 8 to 35 in order to cover the FSF adequately.

### B.3   ANMS-based Keyframe Selection

As we record calibration video, we move the board between intended poses in a jerky manner. This results in sharp frames at the intended key poses with a high number of detected tag corners, and blurry intermediate frames with few corner detections.

We use a variant of adaptive non-maximum suppression (ANMS) [4] to automate keyframe selection. We measure the strength of a frame by the number of tag corners detected, and we define the distance between two frames $F_i$ and $F_j$ as the absolute difference between their indices $|i - j|$. We use robustness parameter of $c_{robust} = 1$, and dynamically select the suppression radius threshold using the elbow method. Note that ANMS will keep all frames with 352 tag corner detections, as this is the maximum possible amount. When consecutive frames all contain 352 corner detections, we assume they correspond to the same pose and retain only the median frame. Our approach reliably selects a high-quality frame for each distinct pose and filters out low-quality and redundant frames, thereby reducing downstream computation costs. See Fig. E for an example of our frame selection.

### B.4   Camera Height

We use a camera height of approximately 1.44 m. At this height, we find that the lower extents of the FSFs begin to clip into the ground before the upuper extents become too high to reasonably hold the board at. When it becomes impossible to place the board at the desired FD while still covering the bottom edge of the camera's FOV, we switch to drone-based calibration instead.

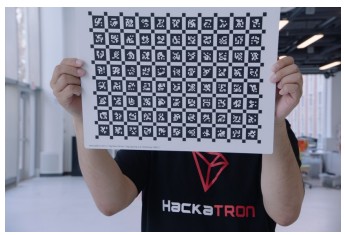

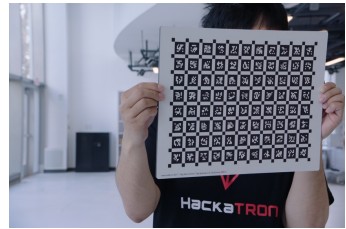 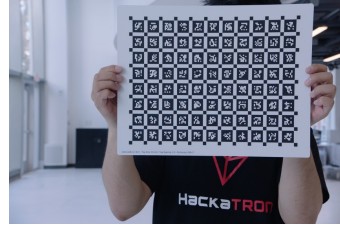 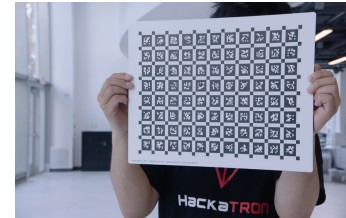

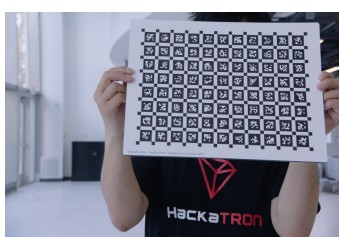

Figure D: The five orientations at which we hold the board at to excite all axes of rotation.

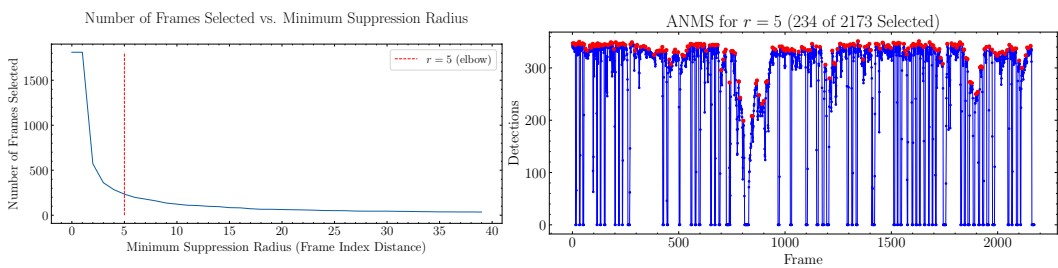

Figure E: The set of selected keyframes using our ANMS-based heuristic. By performing ANMS based on number of tag corner detections and frame index, we are able to automatically choose high-quality frames for each distinct pose and filter out low-quality and redundant frames.

## C  Additional Details for Drone-Based Experiments

### C.1  Drone Hardware

We use a Holybro X500 V2 quadcopter drone with Pixhawk 6X flight controller, M10 GPS, and 915 MHz SiK telemetry radio. We power the drone with an AMORIL 14.8V 4-cell (4s) LiPo battery with a capacity of 5000mAh and a 50C continuous discharge rating.

To enable accurate 2D detections, we use a CHANZON 3 W red LED chip with a forward voltage of 2.0-2.4V and a forward current of 400-500mA. To supply this LED with power, we connect it to the drone's main 14.8V 4S LiPo battery via an XT30 connection onboard the drone. We use a HUABAN buck converter configured in constant current mode to step the voltage down from 14.8V to approximately 2.2V and regulate output current to 450mA, ensuring stable LED operation. Power delivery to the converter is toggled by a HiLetgo 5V single-channel relay module with optoisolation,

which is controlled by our Raspberry Pi (RPi) via general purpose input/output pins (GPIO), enabling us to switch the LED on and off programmatically.

To track the drone's 3D position with high accuracy, we use the Septentrio Mosaic X5 real-time kinematics (RTK) chip. In our setup, the RTK obtains correction data from Point One's Polaris RTK service via a Networked Transport of RTCM via Internet Protocol (NTRIP) stream. Receiving NTRIP data requires internet connection, so we provide connection to the RTK chip via internet over USB using our RPi, which itself is connected to a mobile phone hotspot. This setup enables the RTK system to achieve centimeter-level 3D positioning accuracy in real time.

The final hardware assembly is as follows. On top of the main platform of the drone, we attach the Pixhawk 6X flight controller and the Septentrio Mosaic X5 RTK chip side by side, and we place the M10 GPS on top of the flight controller. On top of the drone's tail platform, we attach the buck converter and single-channel relay module. On the bottom of the drone's tail, we attach our RPi, which is powered via a portable charger power bank that is carried as payload. We craft a cardboard box filled with packing peanuts to carry the main drone battery as well as the phone providing hotspot internet to the RPi, and we add this box to the drone payload as well. We attach our red LED light to the bottom of this payload, and cover the box with matte sandpaper material to reduce the amount of scattered light that is picked up by the camera. See Fig. F for a visualization.

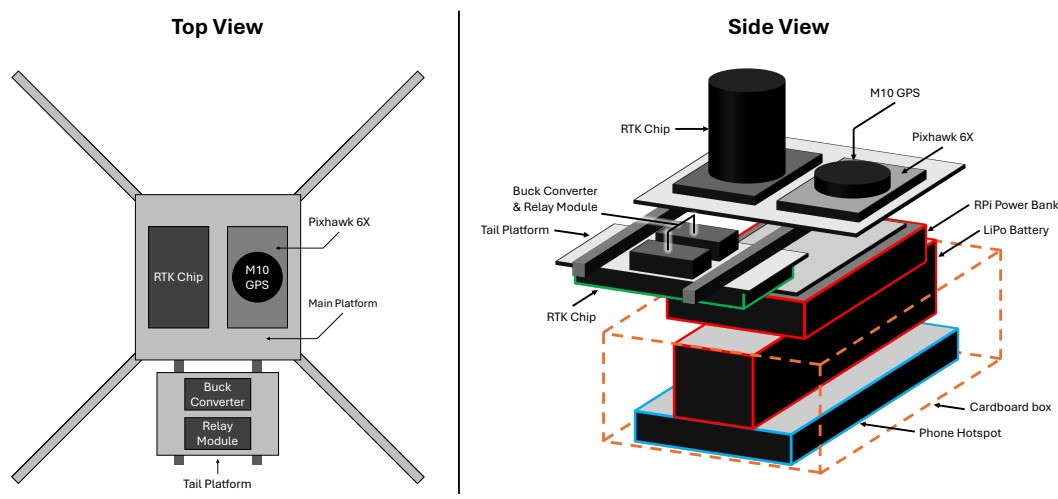

Figure F: Top and side views of our drone hardware setup.

## C.2 Drone RPi Software

During drone calibration, we aim to record data only when the drone is hovering still. To achieve this, we establish a TCP/IP communication link between the drone's onboard RPi and a ground-based computer. On the drone, we run a server that receives a continuous stream of RTK data and controls when the red LED is turned on or off. On the ground, we run a client program which allows us to manually send a trigger signal to the drone over TCP/IP. When the server receives this signal, it turns on the LED, records the current RTK position, turns off the LED after 0.25 seconds, and sends the recorded RTK data back to the client. Upon receiving this data, the client converts the recorded geodetic position to Cartesian coordinates and stores it.

To determine when to send these manual signals, we monitor the drone's position in QGroundControl. When the drone appears to be hovering still, we manually trigger the signal. This process ensures that we capture 3D RTK positions and corresponding 2D LED positions only when the drone is hovering still, thereby establishing reliable 2D–3D correspondences for downstream camera calibration.

## C.3 Camera Space to Geodetic Coordinates Conversion

To be able to fly our drone autonomously, we must compute the transformation from camera space coordinates to geodetic coordinates, as QGroundControl only accepts geodetic coordinates as input for flight planning. We define camera space to have its origin at the camera center, $+x$ pointing towards the right of the image, $+y$ pointing towards the bottom of the image, $+z$ pointing forward along the camera's optical axis, and a scale where 1 unit equals 1 meter. Geodetic coordinates are in the format of (latitude, longitude, altitude). To facilitate computations, we define an intermediate world space, which has origin at the camera center, $+x$ pointing towards east, $+y$ pointing towards north, $+z$ pointing up, and a scale where 1 unit equals 1 meter.

To compute the transformation from camera space to world space, we first measure the camera's latitude $lat_0$ and longitude $long_0$ positions, its height above ground $h$, its compass orientation $\alpha$ defined as counterclockwise rotation from north, and its deviation from normal angle $\theta$ defined as the angle formed between the camera's optical axis and the vertical axis. Then, the following matrix will transform points $p$ in camera space to world space:

$$T_{cam\_to\_world}(p) = \begin{bmatrix} \cos\alpha & \sin\alpha & 0 & 0 \\ -\sin\alpha & \cos\alpha & 0 & 0 \\ 0 & 0 & 1 & 0 \\ 0 & 0 & 0 & 1 \end{bmatrix} \cdot \begin{bmatrix} 1 & 0 & 0 & 0 \\ 0 & \cos\theta & \sin\theta & 0 \\ 0 & -\sin\theta & \cos\theta & h \\ 0 & 0 & 0 & 1 \end{bmatrix} \cdot p$$

To transform from world space to geodetic coordinates, we utilize the python library `pyproj` to do so. It takes in the starting latitude and longitude $(lat_0, long_0)$ and a displacement vector, which in this case is the world space coordinate. It then utilizes azimuthal equidistant projection to compute the final latitudes and longitudes corresponding to world space points. Hence, the overall transform from camera space to geodetic coordinates can be expressed as:

$$T_{cam\_to\_geodetic}(p) = \text{pyproj}(T_{cam\_to\_world}(p), lat_0, long_0)$$

## C.4 3D Flight Path Computation

To compute our drone flightpath, we first need to calibrate the size of the FSF that the flight needs to cover. Given an input LFL and FD setting we wish to calibrate at, we first estimate the value of the CFL in metric units via the thin lens equation as follows:

$$\frac{1}{LFL} = \frac{1}{CFL} + \frac{1}{LTO}$$
$$\frac{1}{LFL} = \frac{1}{CFL} + \frac{1}{FD - CFL}$$
$$\frac{1}{LFL} = \frac{FD}{CFL(FD - CFL)}$$
$$FD \cdot LFL = -CFL^2 + FD \cdot CFL$$
$$0 = CFL^2 - FD \cdot CFL + FD \cdot LFL$$
$$CFL = \frac{FD \pm \sqrt{FD^2 - 4 \cdot FD \cdot LFL}}{2}$$

Because CFL is usually much less than FD, we select the root

$$CFL = \frac{FD - \sqrt{FD^2 - 4 \cdot FD \cdot LFL}}{2} \tag{1}$$

Using this estimate $CFL_{thin}$ and the size of the camera sensor $H \times W$, all in metric units, we can now estimate the FSF. In camera space, we have that the FSF is the region contained within a rectangular plane with corners at

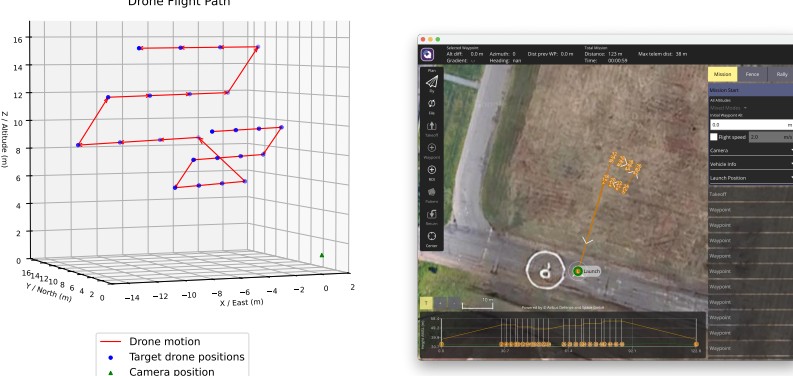

Figure G: A visualization of an example 3D drone path. The camera space coordinates are shown on the left, and the corresponding QGroundControl mission is shown on the right.

$$\left( \pm \frac{sW}{2}, \pm \frac{sH}{2}, s \cdot CFL_{thin} \right), \quad \text{where scale factor } s = \frac{FD - CFL_{thin}}{CFL_{thin}}$$

Because real lenses often deviate from the thin lens model, our initial FSF estimate may be inaccurate. We empirically test our FSF estimate by flying the drone to the geodetic locations corresponding to

$$\left( \pm \left( \frac{sW}{2} - m \right), \pm \left( \frac{sH}{2} - m \right), s \cdot CFL_{thin} \right)$$

in camera space, where $m$ is a margin of 20 cm. If the drone does not appear at the edges of the camera's FOV during this test flight, we manually adjust the value of $CFL_{thin}$ and repeat the test flight. We continue tuning this value until we have found a $CFL^*_{thin}$ that allows the drone to reliably fly to the edges of the camera FOV.

Once the FSF has been tuned, we define our drone's flightpath to be the following set of 24 points:

$$\left( \frac{x - 1.5}{1.5} \left( \frac{psW}{2} - m \right), (y - 1) \left( \frac{psH}{2} - m \right), ps \cdot CFL^*_{thin} \right), \quad \text{where } s = \frac{FD - CFL^*_{thin}}{CFL^*_{thin}},$$

(2)

$$\text{where } m = 20 \text{ cm,}$$
$$\text{for } x \in \{0, 1, 2, 3\},$$
$$\text{for } y \in \{0, 1, 2\},$$
$$\text{for } p \in \{0.75, 1.25\}$$

This is a grid of $4 \times 3 \times 2$ points. We pick two different depths around the LTO, and at each depth, we create a grid of $4 \times 3$ points that spans the FSF at that depth. This is meant to simulate how the set of all tag corners looks like across all board calibration images: a collection of dense planar grids of points at various different depths due to variation in board orientation relative to camera. See Fig. G for a visualization of a computed 3D drone path.

## C.5 2D Keypoint Detection

We record drone calibration footage at night to ensure that the red LED on the drone is highly salient against the night sky. When the LED is flashed on, it will generally appear in the image either as a very intense white ellipse surrounded by a red halo, or an intense red light. We design a set of heuristics around this observation to semi-automate the process of 2D keypoint detection.

Given an input frame, we first look for extremely bright pixels surrounded by a red halo to be candidates for LED position. Specifically, we convert each pixel from RGB to HSL color space, and select all pixels that have a lightness value above 210. We then fit elliptical contours via OpenCV [2], and for each returned ellipse with major and minor radii of $r_{major}$ and $r_{minor}$, we only consider it as a LED position candidate if $r_{minor} > 10$ pixels, $\frac{r_{major}}{r_{minor}} < 1.5$, and if 25% or more of the crop of $2 \cdot r_{major} \times 2 \cdot r_{major}$ around the ellipse center contains red pixels. Red pixels are determined via OpenCV's `cv2.inRange` function, and we select pixels whose HSV values are either between $(0, 100, 120)$ and $(50, 255, 255)$ for the red color, or between $(130, 100, 120)$ and $(180, 255, 255)$ for the magenta color. If any ellipses satisfy all these checks, we select the ellipse whose patch has the highest average lightness value.

If no ellipses are selected by the first heuristic, then we repeat our first heuristic but with slightly relaxed conditions. First, we identify pixels that have a lightness value of 150 rather than 210. We perform elliptical contour fitting as usual, but we only make sure that the ellipses have $r_{minor} > 8$ pixels instead, and drop the $\frac{r_{major}}{r_{minor}}$ check. The red halo check and best ellipse selection is as before.

If no ellipses have been selected still, then we apply a different heuristic looking for intensely red pixels. First, we select all red pixels using `cv2.inRange` as mentioned before. We fit elliptical contours on these selected pixels, and we keep the ellipses that have $r_{minor} > 8$ pixels. Finally, we select the ellipse whose patch has the highest mean saturation, but for two patches with similar saturation within a factor of 0.9, we select the patch that is more circular by comparing $\frac{r_{major}}{r_{minor}}$ values. This modified ellipse selection helps distinguish between the appearance of the LED versus scattering effects, which typically appear to be more elliptical.

After running our heuristic for 2D point detection on every video frame, we expect to have the same number of 2D points and recorded 3D RTK points. If there is a mismatch, then we manually review the video clip and determine the center of the LED if it was not successfully detected by the heuristics.

# D  Additional Details for Kalibr Modifications

As mentioned in the main paper, Kalibr's calibration algorithm occurs in four stages: (1) camera pose initialization via perspective-n-point; (2) CFL initialization using vanishing points (VP) [7]; (3) distortion initialization via Levenberg–Marquardt (LM); and (4) joint optimization of poses and intrinsics using LM. Our modified Kalibr differs from original Kalibr for steps 2, 3, and 4. Importantly, to process drone calibration data, we also modify Kalibr to support 2D-3D point correspondence input for any 3D target, a relaxation of original Kalibr's constraint of only working with known planar calibration targets.

**Modified CFL Initialization.**  Instead of relying on the VP-based algorithm in [8] to initialize the CFL, we instead estimate the CFL based on the thin lens model. To do so, we plug in the experiment's LFL and FD values into Eq. (1). This estimate is in metric units, so we convert to units of pixels by dividing by pixel size. For our ARRI Alexa Mini, its sensor size is 28.25 mm $\times$ 18.17 mm, and its image resolution for ARRIRAW format is $3424 \times 2202$ pixels. Hence, we use a pixel size of approximately 0.00825 mm.

**Modified Distortion Initialization via Fixed Point Algorithm.**  The original version of Kalibr runs LM algorithm once to optimize the CFL, principal point, and distortion parameters simultaneously. However, because CFL initialization is not guaranteed to be accurate, the initialization point for this step's LM algorithm may lead to inaccurate results. When the output is inaccurate, we observe that the principal point often drifts far from the image center as well.

To address this, we propose our fixed point initialization algorithm that aims to: (1) use the inductive bias that our camera system's principal point should be close to the image center; and (2) ensure that the initialization of any LM run is as accurate as we can make it. Given an initial CFL estimate $CFL_{init}$ in pixels as well as the image dimensions $H \times W$, our goal is to initialize a set of eight intrinsics values $(f_x, f_y, c_x, c_y, k_1, k_2, p_1, p_2)$, where $f_x$ and $f_y$ are CFL in pixels, $c_x$ and $c_y$ denote the principal point location in pixels, and $k_1, k_2, p_1, p_2$ are distortion parameters following the Brown-Conrady distortion model [3]. Let $outputs \leftarrow LM(\dots)$ denote running the LM algorithm, which takes in as input $(f_x, f_y, c_x, c_y, k_1, k_2, p_1, p_2)$ and outputs a set of predictions $(f_{[x,LM]}, f_{[y,LM]}, c_{[x,LM]}, c_{[y,LM]}, k_{[1,LM]}, k_{[2,LM]}, p_{[1,LM]}, p_{[2,LM]})$. The algorithm is as follows:

**Algorithm 1** Distortion Initialization via Fixed Point Algorithm

---

1: **procedure** FIXED POINT INITIALIZATION($CFL_{init}$, $H$, $W$)
2:     $(f_x, f_y, c_x, c_y, k_1, k_2, p_1, p_2) \leftarrow (CFL_{init}, CFL_{init}, \frac{W}{2}, \frac{H}{2}, 0, 0, 0, 0)$
3:     **for** $i = 1$ to 3 **do**               ▷ Refine CFL estimate first without distortion
4:         $outputs \leftarrow LM(\dots)$
5:         $(f_x, f_y) \leftarrow (\frac{f_{[x,LM]} + f_{[y,LM]}}{2}, \frac{f_{[x,LM]} + f_{[y,LM]}}{2})$         ▷ Average CFL predictions
6:         $(c_x, c_y) \leftarrow (\frac{W}{2}, \frac{H}{2})$                ▷ Snap principal point to center
7:         $(k_1, k_2, p_1, p_2) \leftarrow (0, 0, 0, 0)$            ▷ Zero-out distortion
8:     **end for**
9:     $(f_x, f_y, c_x, c_y, k_1, k_2, p_1, p_2) \leftarrow LM(\dots)$     ▷ Extra optimization steps (optional)
10:    **for** $i = 1$ to 4 **do**             ▷ Jointly refine CFL and distortion estimate
11:        $outputs \leftarrow LM(\dots)$
12:        $(f_x, f_y) \leftarrow (\frac{f_{[x,LM]} + f_{[y,LM]}}{2}, \frac{f_{[x,LM]} + f_{[y,LM]}}{2})$        ▷ Average CFL predictions
13:        $(c_x, c_y) \leftarrow (\frac{W}{2}, \frac{H}{2})$              ▷ Snap principal point to center
14:    **end for**
15:    **return** $(f_x, f_y, c_x, c_y, k_1, k_2, p_1, p_2)$
16: **end procedure**

---

**Modified Joint Optimization of Poses and Intrinsics.** The joint optimization process for original Kalibr is stochastic because it processes all frames one by one in random order for outlier rejection. Empirically, we observe that some random frame orderings lead to outlier predictions of intrinsics, compared to other runs of Kalibr with different frame ordering. To reduce the variance of Kalibr, we modify it so that we actually perform 100 separate rollouts of the joint optimization step.

From these 100 rollouts, we pick one representative rollout as follows. Across all 17 rollouts' $f_x$ values, we identify outlier rollouts by using the interquartile range (IQR) method. Similarly, we identify outlier rollouts over $f_y$, $c_x$, and $c_y$ values. Next, we discard rollouts that were outliers over any of $f_x$, $f_y$, $c_x$, or $c_y$. Among the remaining rollouts, we compute the median values of non-distortion intrinsics; denote these as $f_{x,med}$, $f_{y,med}$, $c_{x,med}$, and $c_{y,med}$. Given a rollout with non-distortion intrinsics predictions of $f_x'$, $f_y'$, $c_x'$, and $c_y'$, we assign it a score of its summed percent deviation from median values:

$$\text{Rollout Score} = \left( \frac{|f_x' - f_{x,med}|}{f_{x,med}} + \frac{|f_y' - f_{y,med}|}{f_{y,med}} + \frac{|c_x' - c_{x,med}|}{c_{x,med}} + \frac{|c_y' - c_{y,med}|}{c_{y,med}} \right) \cdot 100$$

We then pick the rollout with the lowest score as the representative rollout for the LFL-FD setting.

In the case that the representative rollout's principal point drifts further than 2% from the image center for either $c_x$ or $c_y$, we choose to revert back to the intrinsics initialization produced at the end of step (3). Specifically, this occurs when $\frac{|c_x - 1712|}{1712} > 2\%$ or $\frac{|c_y - 1101|}{1101} > 2\%$, as our image center is $(1712, 1101)$. This guarantees that the principal point is at the center of the image, and the CFL approximation at this stage is also empirically close to ground truth in our synthetic experiments.

**Inputting General 2D-3D Correspondences.** We modify Kalibr to accept 2D-3D point correspondences with respect to arbitrary 3D target. The 3D structure of the target is specified by providing Kalibr a list of 3D points that are normalized to fit within a unit sphere. Corresponding 2D detections are passed in as a list of $T \cdot N$ 2D points, where $T$ is the number of points in the 3D target and $N$ is the number of frames, where each frame corresponds to a unique camera viewpoint of the target. The ordering of the $T$ 2D points within a frame corresponds to the ordering of the $T$ points in the 3D target. In addition, missing detections can be indicated via a boolean validity flag for each 2D point. For our drone experiments, because we keep our camera still relative to the scene, we only use one frame in total for 2D-3D correspondence specification.

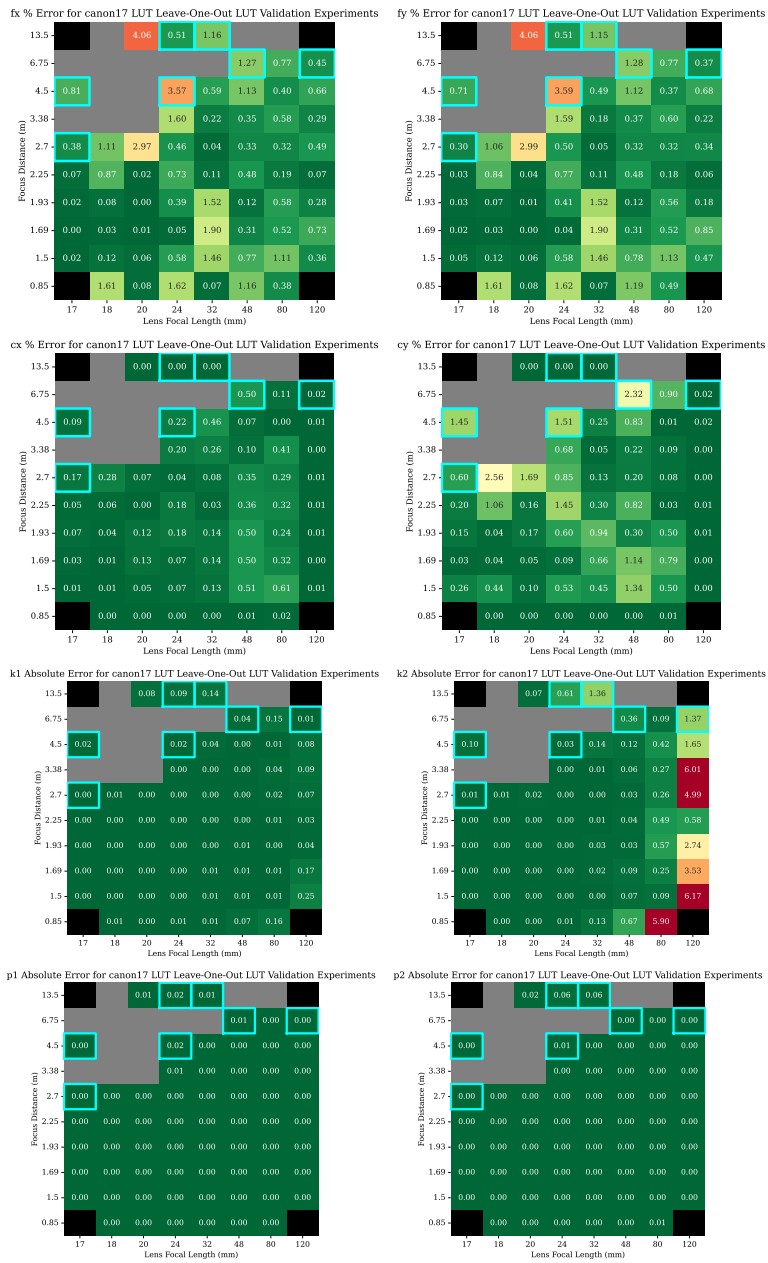

Figure H: LUT cross-validation results for canon17 lens. We report percent error for CFL and principal point parameters, and absolute error for distortion parameters. Datapoints outlined in teal use triangular interpolation; all other datapoints use trapezoidal bilinear interpolation.

# E   Additional Details for LUT Interpolation Scheme

For query points within the LUT bounds, we apply trapezoidal bilinear interpolation over approximately regular grids in LFL-FD space and barycentric interpolation over non-grid regions. Details of the trapezoidal bilinear interpolation are provided in § E.1. Barycentric interpolation is performed using the standard method. We validate the accuracy of our interpolation scheme through leave-one-out cross-validation on calibration data in § E.2. For query points outside the LUT bounds, we do not provide interpolation results for evaluation purposes. However, we provide a best-guess extrapolation following a thin lens-inspired model as described in § E.3.

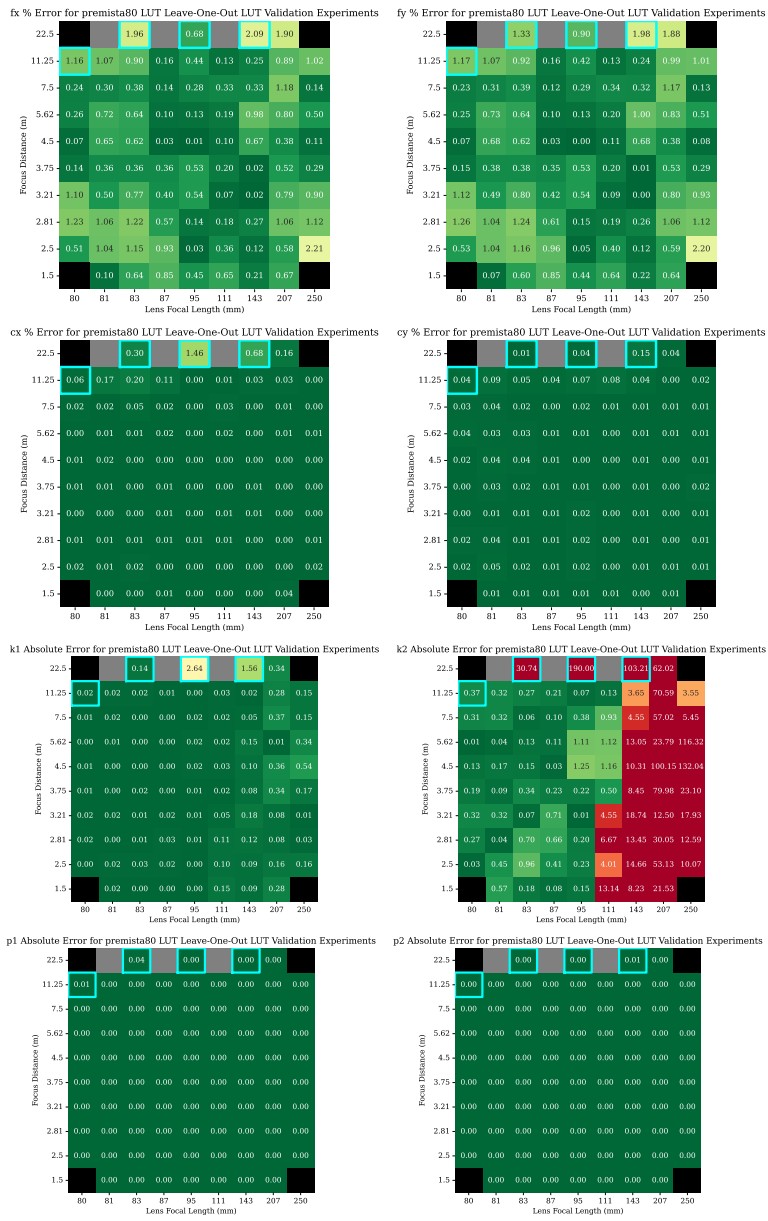

Figure I: LUT cross-validation results for premista80 lens. We report percent error for CFL and principal point parameters, and absolute error for distortion parameters. Datapoints outlined in teal use triangular interpolation; all other datapoints use trapezoidal bilinear interpolation.

## E.1 Trapezoidal Bilinear Interpolation

When configuring our lens for certain experiment settings, LFL can be set precisely, but FD cannot be set precisely because the camera's UI display of FD has less precision than what is actually recorded in lens metadata. Hence, the grid-like regions in our LUT are trapezoidal rather than rectangular. Consider a trapezoidal region enclosed by the points $A : (LFL_0, FD_0)$, $B : (LFL_0, FD_1)$, $C : (LFL_1, FD_2)$, and $D : (LFL_1, FD_3)$, where $LFL_0 < LFL_1$ and $FD_0 \approx FD_3 < FD_1 \approx FD_2$. Let the values associated with the four points be $a$, $b$, $c$, and $d$, respectively. We will now show how to do trapezoidal bilinear interpolation for a query point $(LFL, FD)$.

Define $P_{LFL} = \frac{LFL - LFL_0}{LFL_1 - LFL_0}$. To define the corresponding fraction for the $FD$ axis, we first compute where the line $x = FD$ intersects $\overline{AD}$ and $\overline{BC}$. The intersections can be computed as

$$\text{Intersection with } \overline{AD}: \quad (LFL, y_1), \quad \text{where } y_1 = \frac{FD_3 - FD_0}{LFL_1 - LFL_0} \cdot (LFL - LFL_0) + FD_0$$

$$\text{Intersection with } \overline{BC}: \quad (LFL, y_2), \quad \text{where } y_2 = \frac{FD_2 - FD_1}{LFL_1 - LFL_0} \cdot (LFL - LFL_0) + FD_1$$

Define $P_{FD} = \frac{FD - y_1}{y_2 - y_1}$. Then, the final trapezoidal bilinear interpolation result is

$$
\begin{aligned}
(1 - P_{LFL}) \cdot (1 - P_{FD}) \cdot a + \\
(1 - P_{LFL}) \cdot P_{FD} \cdot b + \\
P_{LFL} \cdot P_{FD} \cdot c + \\
P_{LFL} \cdot (1 - P_{FD}) \cdot d
\end{aligned}
$$

### E.2 LUT Cross-Validation Experiment Results

We conduct leave-one-out cross-validation experiments which show that the LUT interpolation is accurate and produces reliable ground truth. In these experiments, we withhold one calibration datapoint from the LUT at a time and evaluate the accuracy of the camera intrinsic estimate for the held-out point by interpolating among its neighboring datapoints. To determine the appropriate interpolation method, we first collect the set of all datapoints in regions involving the input datapoint. We then remove the input datapoint and check whether the remaining datapoints form an axis-aligned grid region. If at least one axis-aligned grid region exists, we take the region with least interpolation over LFL and apply trapezoidal bilinear interpolation. Otherwise, we apply Delaunay triangulation over the remaining datapoints, select the triangle that contains the withheld input datapoint, and perform triangular interpolation. Across both lenses, the median CFL error is $< 0.5\%$ (max error $< 4.1\%$), and the median principal point error is $< 0.2\%$ (max error $< 2.6\%$). We report the absolute error for distortion parameters. Overall, the accuracy of each intrinsic is good for both lenses. The highest absolute errors occur for $k_2$ interpolation, but this distortion parameter only produces very minor fringe distortion effects and is relatively negligible. See Figs. H and I for results.

### E.3 LUT Extrapolation Scheme

For points outside our LUT, we cannot apply interpolation to retrieve intrinsics. However, we can produce a best-guess extrapolation in the following manner. Consider a query point of $(LFL_q, FD_q)$ outside the bounds of our LUT. By nature of the lens metadata collected, the LFL value will always be within the bounds of the LUT, but the FD will be greater than the maximum FD the LUT supports.

For all intrinsics other than CFL, we snap the query point's FD value to the nearest in-bounds FD and report that point's interpolated intrinsic values. For extrapolating the CFL values $f_x$ and $f_y$, we utilize intuition from the thin lens model. Let $LFL_0$ and $LFL_1$ denote the two LFL values closest to $LFL_q$ for which we have calibration experiments for, such that $LFL_0 \leq LFL_q \leq LFL_1$. Let $S_0 = \{(LFL_0, FD_i)\}_{0 \leq i < k}$ denote the set of all $k$ calibration experiments the LUT contains for $FD_0$, and let $f_{x,i}$ and $f_{y,i}$ denote each experiment's $f_x$ and $f_y$ values, respectively. We can compute the approximate CFL for each experiment in metric units by scaling both $f_{x,i}$ and $f_{y,i}$ by pixel size and averaging:

$$CFL_{metric,i} = 0.5 \cdot \left[ f_{x,i} \cdot \frac{\text{sensor width mm}}{\text{sensor resolution x}} + f_{y,i} \cdot \frac{\text{sensor height mm}}{\text{sensor resolution y}} \right]$$

The thin lens equation states that for an ideal thin lens, we have that

$$\frac{1}{LFL} = \frac{1}{CFL} + \frac{1}{FD - CFL}$$

Then, we can compute the approximate thin lens effective LFL for $LFL_0$ by averaging $\frac{1}{CFL} + \frac{1}{FD - CFL}$ over all experiments and taking the reciprocal:

$$LFL_{0,eff} = \frac{k}{\sum_{i=0}^{k-1} \left[ \frac{1}{CFL_{metric,i}} + \frac{1}{FD_i - CFL_{metric,i}} \right]}$$

Using this value, we can compute an extrapolated CFL value $CFL_{q,0}$ at $(LFL_0, FD_q)$ via thin lens equation approximation as derived in Eq. (1):

$$CFL_{q,0} = \frac{FD_q - \sqrt{FD_q^2 - 4 \cdot FD_q \cdot LFL_{0,eff}}}{2}$$

We can similarly compute an extrapolated CFL value $CFL_{q,1}$ at $(LFL_1, FD_q)$. Finally, we can linearly interpolate over LFL value between these two approximations via

$$\left( 1 - \frac{LFL_q - LFL_0}{LFL_1 - LFL_0} \right) \cdot CFL_{q,0} + \left( \frac{LFL_q - LFL_0}{LFL_1 - LFL_0} \right) \cdot CFL_{q,1}$$

We report this extrapolated value for both $f_x$ and $f_y$.

## F  Additional Details for Kalibr Synthetic Experiments Design

To evaluate the performance of our modified Kalibr algorithm versus the original Kalibr, we design a set of calibration experiments with settings similar to the distribution of those used in real-world LUT experiments. For each lens, we use the same LFL and FD settings as those used in real-world experiments. From this, we utilize Eq. (1) to compute the corresponding CFL under the thin-lens model, and we use this CFL as ground truth when rendering synthetic images of calibration targets. We now describe how we choose the amount of distortion to add to each experiment, which calibration target to use, and how we render synthetic input images for each experiment.

**Synthetic Distortion.** For our experiments, we choose the amount of distortion to add such that the overall range of synthetic distortion encompasses the range of distortion observed in real-world LUT calibration experiments. Given an undistorted image coordinate $(x, y)$ with CFL $f$, and assuming the image center is $(0, 0)$, we first need to normalize the image coordinates: $x_{norm} = \frac{x}{f}$ and $y_{norm} = \frac{y}{f}$. Then, the Brown-Conrady distortion model [3] maps the coordinate to

$$x_{distort} = x_{norm}(1 + k_1 r^2 + k_2 r^4 + k_3 r^6) + 2p_1 x_{norm} y_{norm} + p_2(r^2 + 2x_{norm}^2)$$
$$y_{distort} = y_{norm}(1 + k_1 r^2 + k_2 r^4 + k_3 r^6) + p_1(r^2 + 2y_{norm}^2) + 2p_2 x_{norm} y_{norm}$$
$$\text{where } r^2 = x_{norm}^2 + y_{norm}^2$$

Note that typically, $r \ll 1$ after normalization, and for high end cameras, there is little to no tangential distortion. Hence, we may approximate the distorted position as

$$x_{distort} \approx x_{norm}(1 + k_1 r^2)$$
$$y_{distort} \approx y_{norm}(1 + k_1 r^2)$$

Using this approximation, we can plug expression back in to derive

$$x_{distort} - x_{norm} = x_{norm} k_1 r^2$$
$$x_{distort} - x_{norm} = \frac{x}{f} k_1 \frac{x^2 + y^2}{f^2}$$
$$f \cdot (x_{distort} - x_{norm}) = k_1 \frac{x(x^2 + y^2)}{f^2}$$

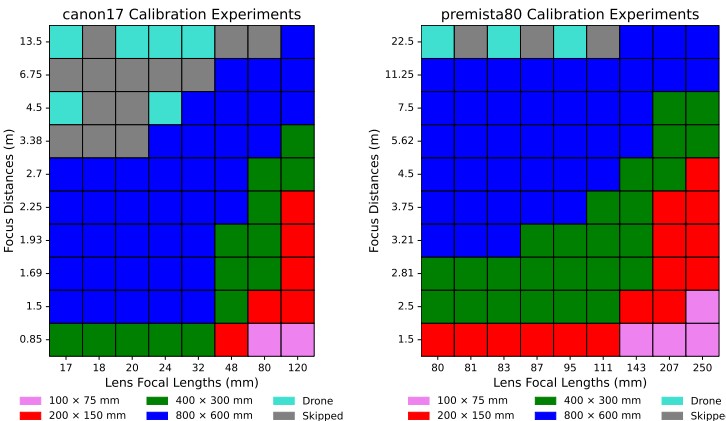

Figure J: A visualization of the different synthetic calibration experiments performed to evaluate Kalibr performance for canon17 and premista80. Our synthetic experiments use the same LFLs and FDs as real-world experiments. The distortion applied is also chosen to cover the range of distortion seen in real-world experiments.

Note that $f \cdot (x_{distort} - x_{norm})$ represents the displacement due to distortion effects, in pixels. For our synthetic experiments, we aim to produce a smooth visual change in the amount of displacement due to distortion as CFL changes. Hence, we need the quantity $\frac{k_1}{f^2}$ to vary smoothly as CFL changes. We define this fraction as the visual factor for displacement caused by distortion.

For each lens, we pick a minimum value of $k_1$ when the CFL is at a minimum to make barrel distortion; denote these as $k_{1,min}$ and $CFL_{min}$. Then, the visual factor is $\frac{k_{1,min}}{CFL_{min}^2}$, and we aim to have the visual factor vary smoothly from $\frac{k_{1,min}}{CFL_{min}^2}$ to $-\frac{k_{1,min}}{CFL_{min}^2}$ as CFL changes. Hence, for a choice of a specific CFL, this results in choosing $k_1$ as follows:

$$k_1, \text{ given choice of CFL:} \quad k_1 = CFL^2 \cdot VF$$
$$\text{where } VF = (1 - 2P) \cdot \frac{k_{1,min}}{CFL_{min}^2}$$
$$\text{where } P = \frac{CFL - CFL_{min}}{CFL_{max} - CFL_{min}}$$

**Calibration Target Choice.** To mirror real-world small to medium FSF experiments, we create virtual calibration boards with the same size and patterning. We choose the largest possible size of calibration board such that the board fits within the camera FSF, even when rotated by 45 degrees to excite axes of rotation.

To simulate real-world settings, we transition to synthetic drone-based calibration when the largest board cannot fit within the camera FSF without clipping into the ground at the set FD. By similar triangles, we use synthetic drone targets instead when the following condition holds:

$$FD > \frac{2 \cdot h_{cam} \cdot CFL}{h_{sensor}}$$

Here, we assume camera height is $h_{cam} = 1.44$ m, and height of the camera sensor is $h_{sensor} = 0.01817$ m. See Fig. J for the choices of calibration target for each synthetic experiment.

**Rendering Synthetic Targets.** For each synthetic calibration board experiment, we use Blender to render a set of 100 images. We set the scale of the scene so that 1 unit of distance is equal to 1 meter. We render the board in each of the five board orientations demonstrated in Fig. D at a 45 degree angle if applicable. For each board orientation, we move it around the camera FSF in a $4 \times 5$ grid of

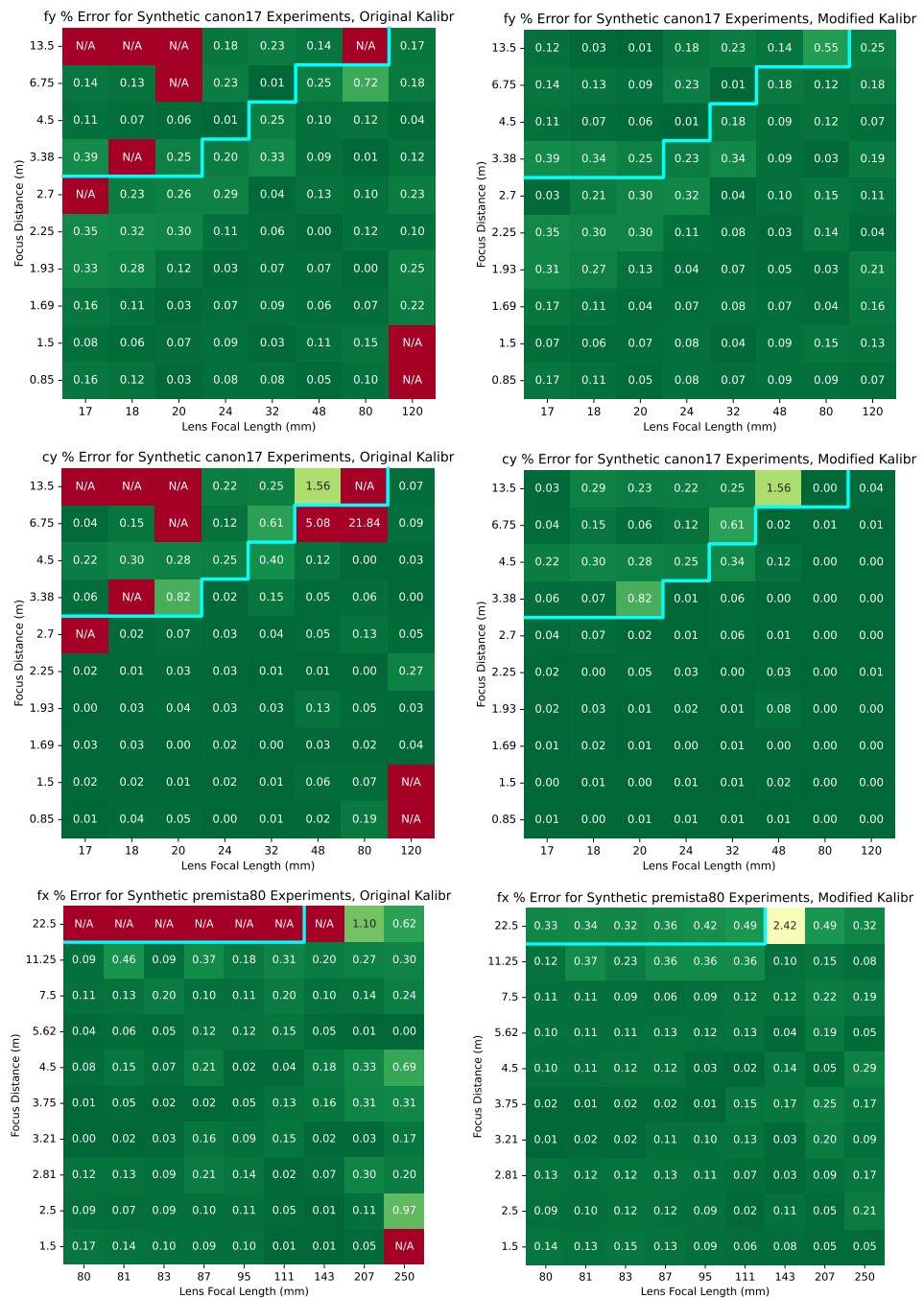

Figure K: Additional synthetic experiment results for old Kalibr (left) versus modified Kalibr (right). Second part of figure on next page.

positions. We feed the 100 rendered images directly to both versions of Kalibr for AprilGrid corner detection and intrinsics prediction.

For each synthetic drone experiment, we use Blender to render a red sphere of radius 0.02 m at the 3D locations corresponding to a real drone flight path defined in Eq. (2). We keep the background white and apply a hue-based filtering with elliptic contour to identify the mock LED center for synthetic 2D detections. For synthetic 3D detections, we take the ground truth positions of

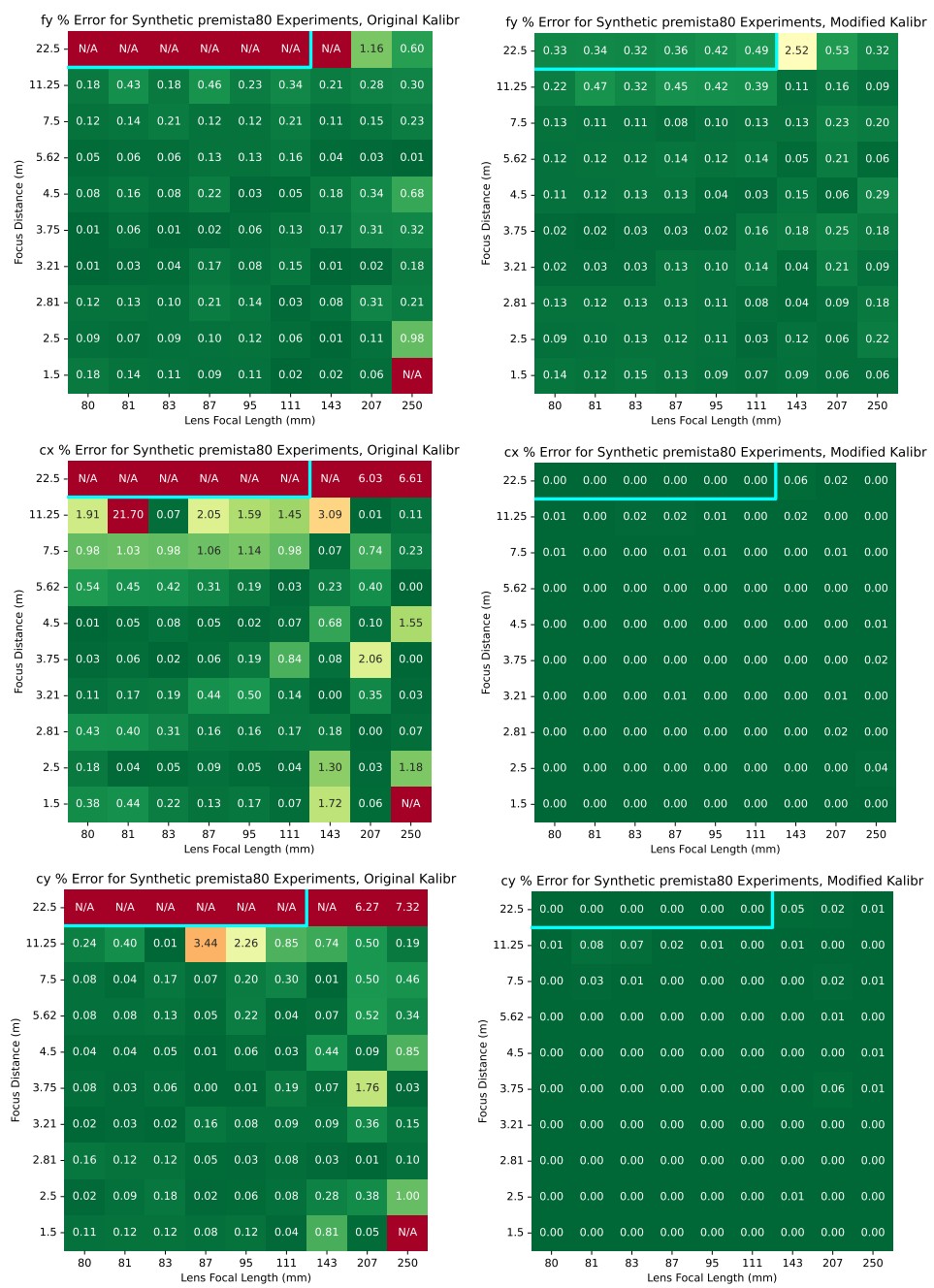

Figure L: Additional synthetic experiment results for old Kalibr (left) versus modified Kalibr (right).

the mock LED and independently perturb each point by adding a random vector sampled uniformly within a sphere of radius 0.5 cm radius to simulate RTK noise.

# G   Additional Evaluation Results for Kalibr Synthetic Experiments

In the main paper, we show a comparison of original Kalibr versus our modified version of Kalibr on synthetic experiments for canon17, in terms of $f_x$ and $c_x$ percent deviation from ground truth. In Figs. K and L, we show additional results for all non-distortion intrinsics over both lenses. From

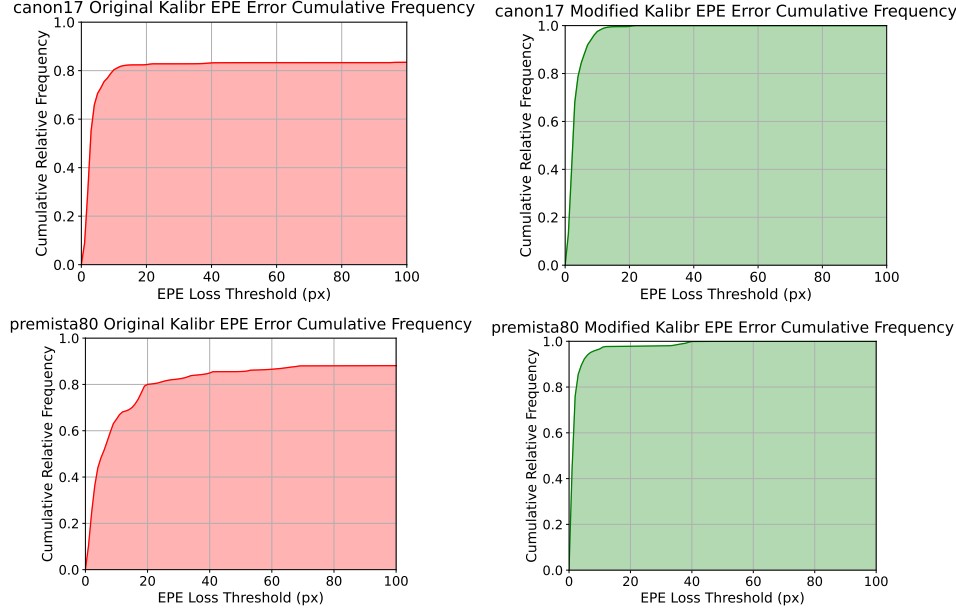

Figure M: Plots of the cumulative relative frequency of EPE errors for original Kalibr and our modified version of Kalibr, evaluated on our synthetic calibration experiments. Overall, our modified version Kalibr is more accurate and reliable compared to the original Kalibr.

these results, we can see that the modified version of Kalibr results in more accurate and consistent results over $f_x$, $f_y$, $c_x$, and $c_y$.

Percent deviation from ground truth is an easy way to measure intrinsics prediction accuracy over non-distortion parameters, but this metric is not appropriate for measuring accuracy of predicted distortion parameters. To evaluate accuracy of intrinsics predictions holistically, we also measure the end point error (EPE) resulting from projecting a set of 100K 3D points with ground truth intrinsics and predicted intrinsics.

**EPE-based Evaluation Details.** For each of [28]'s high resolution scenes, we take the 3D pointcloud of the first scan and define a custom camera position and orientation within the scene to capture subjects of interest within the camera FOV. Based on the set of all camera intrinsics over our real-world benchmark videos, we filter out points that are never in FOV and sample 100K points randomly from the remaining points. We fix this set of points, which serves as a representation of naturally occurring 3D structure in both indoor and outdoor scenes.

For baseline evaluations in the main paper, consider a single frame from our real-world benchmark. Using the frame's ground truth intrinsics derived from our LUTs, we select the subset of the 100K chosen points that fit within FOV. We then project this subset of points onto the camera sensor using both ground truth intrinsics and the baseline method's predictions. For each corresponding pair of projections, we measure the $L_2$ distance between them. If the baseline fails to predict intrinsics, the EPE is considered to be infinite. We analyze the aggregate EPE values over all frame-point pairs.

We can also apply this analysis to our synthetic experiments by considering each LFL-FD setting to be its own frame with associated ground truth intrinsics. See Fig. M for plots. From the plots, we can see that original Kalibr fails to converge on some experiments whereas our version of Kalibr always converges. The intrinsics predicted by our version of Kalibr are also much more accurate compared that those of original Kalibr.

## H  Additional Details for Real-World Evaluation

**Validation and Test Split.** We divide our real-world benchmark videos into separate validation and test splits, ensuring that each video belongs entirely to one split. The validation set comprises approximately $15\%$ of all frames. For validation frames, we provide three types of annotations:

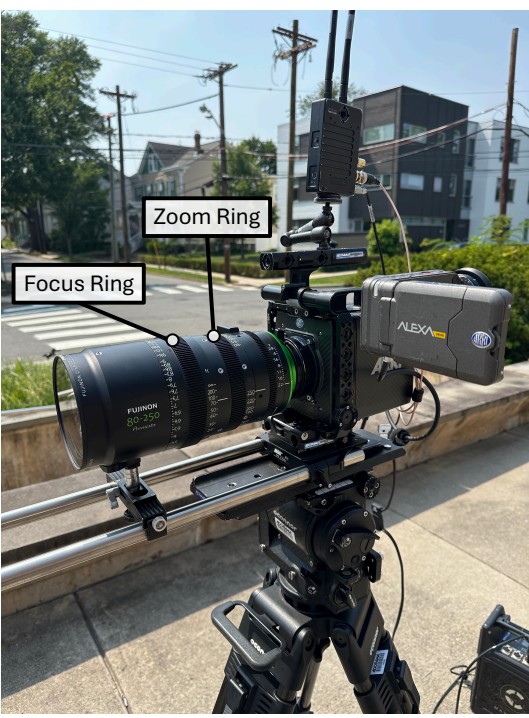

Figure N: Diagram of the premista80 lens used. Rotating its zoom and focus rings shifts internal lens groups, changing the camera intrinsics.

ground truth intrinsics without extrapolation, extrapolated intrinsics, and raw lens metadata. In the test split, evaluation is conducted only on frames with valid ground truth intrinsics, without extrapolation.

# I  Computation Resources Used

For all experiments, we utilize a single GeForce RTX 4090 GPU. Processing one synthetic LFL-FD experiment with our version of Kalibr takes roughly 10 to 15 minutes to run. For real experiments, runtimes may be longer due to extra data preprocessing steps and frame selection. Shorter calibration videos take around 20 to 30 minutes to run, while longer calibration videos can take up to an hour.

# J  Real Life Example of Zoom and Focus Rings

The zoom and focus rings on a lens physically move internal lens groups, which alters the zoom and focus distance and thereby changes the camera intrinsics. See Fig. N for a diagram of the rings on the premista80 lens we use.

# K  Future Works

One direction of future work is to leverage the validation set of our benchmark to train better models for predicting camera intrinsics. If additional data is needed for training purposes, our calibration pipeline and modified Kalibr algorithm can be used to expand the benchmark to more scenes, cameras, and lens types that support lens metadata protocols such as /i Technology.

Another direction for future work is to develop models that jointly predict camera intrinsics and downstream dense vision tasks that depend on accurate intrinsics. One such task is monocular depth estimation, which heavily relies on precise intrinsics; see Fig. O for an illustration of this dependency. Our validation split can be used to train or fine-tune the intrinsics prediction branch of such models.

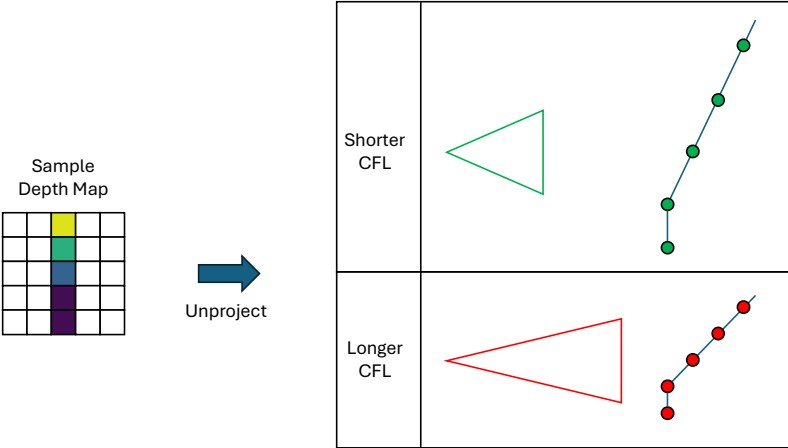

Figure O: An illustration of how camera intrinsics can greatly impact the 3D structure of a scene in the context of monocular depth prediction. Given a fixed depth map and camera position, varying the CFL can distort the relative angle between surfaces and apparent scene geometry.

## L    Societal Impacts

InFlux serves as an evaluation benchmark for understanding dynamic camera intrinsics in video. This may encourage further progress towards making 3D algorithms more robust to in-the-wild videos that have dynamic camera intrinsics. While this can be useful, this could also lead to unwanted or unauthorized 3D reconstruction and scene understanding of videos found online.

## M    Benchmark and Code Licensing

We publicly release two components of our work: the real-world video benchmark, and the code used to run synthetic experiments, process real-world data, and generate ground-truth intrinsics programmatically. These components are licensed separately:

- Code: Released under the BSD 3-Clause license, allowing free use, modification, and redistribution.
- Real-World Video Benchmark: Released under Creative Commons Attribution 4.0 (CC BY 4.0). Users may freely use, share, and adapt the dataset, provided that appropriate credit is given and this work is cited in publications.

