# OpenReview forum: "InFlux: A Benchmark for Self-Calibration of Dynamic Intrinsics of Video Cameras"
_NeurIPS.cc/2025/Datasets_and_Benchmarks_Track — NeurIPS 2025 Datasets and Benchmarks Track poster_

### Official Review · Reviewer_mapJ · 2025-06-11

**Rating:** 5
**Confidence:** 5

**Summary:**

This paper addresses the underexplored but critical problem of dynamic camera intrinsics in real-world video. The authors present ChangeIn, a comprehensive benchmark dataset containing 143K+ frames across 386 high-resolution videos, with per-frame ground truth intrinsics. The benchmark is created using a thoughtful calibration pipeline, combining board-based and drone-based calibration experiments, and an extended version of Kalibr for robustness. Lookup tables (LUTs) indexed by lens focal length (LFL) and focus distance (FD) enable scalable, per-frame intrinsic retrieval. The work is well-motivated, methodologically sound, and fills a crucial gap by providing high-quality, realistic data for evaluating self-calibration methods.

**Dataset Code Accessibility:**

Yes

**Dataset Code Comments:**

Clear and comprehensive codes.

**Ethical Considerations:**

No, there are no or only very minor ethics concerns

**Final Justification:**

I have no major concerns about the work, and the author has addressed my questions adequetely.

**Limitations Weaknesses:**

1. Assumption on Sufficiency of LFL and FD
   The paper assumes that camera intrinsics can be fully determined by the combination of lens focal length (LFL) and focus distance (FD). While this is a practical and solid assumption, it could be explained more clearly. E.g., a visual figure showing how these parameters are adjusted on a real lens (e.g., zoom ring and focus ring) would make the assumption more intuitive and concrete for readers.

2. Missing Related Work on Learning-Based Calibration and Monocular Metric Depth Estimation
   The paper omits discussion of several recent learning-based camera calibration methods that also aim to handle unconstrained or dynamic intrinsics. Notable omissions include:
   - Perspective Fields for Single Image Camera Calibration (CVPR 2023)
   - Tame a Wild Camera: In-the-Wild Monocular Camera Calibration (NeurIPS 2023)
   - UniDepth: Universal Monocular Metric Depth Estimation (CVPR 2024) [It uses a pixel-wise camera model]

   Including these would help position the work within the broader self-calibration literature. It would also help by highlighting the important relationship between intrinsic estimation and monocular metric depth estimation, where accurate intrinsics are essential for recovering scale-consistent 3D structure.

**Strengths Contributions:**

1. Important Problem: The work addresses a critical gap in existing datasets—the lack of videos with dynamic intrinsic changes.

2. Comprehensive Calibration Pipeline: The combination of board-based and drone-based calibration, paired with LUT construction and Kalibr modifications, is technically strong and well-documented.

3. High-Quality Dataset: The benchmark includes a large number of diverse, realistic videos with varying camera motion and intrinsic changes, making it valuable for the community.

4. Clear Impact: The benchmark will likely drive future work in self-calibration, SfM/SLAM/Monocular-Depth-Estimation under dynamic optics, and related downstream tasks.

---

> ### Author Rebuttal · Authors · 2025-07-31
>
> Thank you for the constructive review. Responses to key points:
>
> ---
> ### Concern 1:
> _The sufficiency of lens focal length (LFL) and focus distance (FD) to determine camera intrinsics can be made more clear and intuitive by adding a visual figure._
>
> #### Response:
> Thank you for pointing this out. We will include **an additional figure of a real lens** showing its zoom and focus rings in the camera-ready version to help make this sufficiency of LFL and FD for camera intrinsics more concrete and intuitive.
>
> ---
> ### Concern 2:
> _There are missing related works for methods that predict intrinsics or monocular metric depth, such as PerspectiveFields, (CVPR 2023) WildCamera (NeurIPS 2023), and UniDepth (CVPR 2024)._
>
> #### Response:
> Although we originally omitted these works because they are either superseded by stronger baselines we already evaluated or do not predict distortion parameters, we will **update our related works** and **run additional baseline evaluations** on the mentioned methods to address this concern.
>
> Here are some preliminary baseline results for these works; they have currently been evaluated on a subset of our benchmark's test split, due to time constraints. Overall, these methods do not perform as well as our other baselines so far.
>
> | Method               | % fx Error | % fy Error    | % cx Error      | % cy Error| % Points < 300 px EPE |
> | :------------------- | :---------:| :------------:| :--------------:| :-------: | :-------------------: |
> | Perspective Fields   |     71.0    |   71.0   |    18.9    |    22.3    |         1.86e-1          |
> | WildCamera           |     47.4    |   49.2   |    3.92    |    10.1    |         7.31e-1          |
> | UniDepth             |     56.9    |   57.0   |    1.83    |    2.51    |         8.35e-1          |
>
> There are several reasons for the worse performance of these baseline methods. **PerspectiveFields is superseded by GeoCalib** (ECCV 2024). **WildCamera and UniDepth** use a pinhole camera model and **do not predict distortion**, which is critical for accurate camera modeling. Our **preliminary evaluation of these methods** on our benchmark shows that these methods have poor performance, and we **will provide full evaluation results and analysis** in the camera-ready version.
>
> ---
> ### Concern 3:
> _The relationship between intrinsics estimation and monocular metric depth estimation can be highlighted more._
>
> #### Response:
> Thank you for pointing this out. We will **add additional discussion and figures** to illustrate the important relationship between accurate intrinsics and monocular metric depth estimation in the context of 3D reconstruction.
>
> ---
>
>
>
> Thank you again for your time and feedback.

---

### Official Review · Reviewer_CvcE · 2025-07-01

**Rating:** 6
**Confidence:** 4

**Summary:**

This paper presents the first work of a large-scale video dataset with per frame camera intrinsics annotation, named ChangeIn. ChangeIn captures a wider range of intrinsic variations and scene diversity, featuring 143K+ annotated frames from 386 high-resolution indoor and outdoor videos with dynamic camera intrinsics. A comprehensive lookup table of calibration experiments and Kalibr toolbox is introduced to ensure accurate per-frame intrinsics. With this benchmark, authors evaluate existing camera intrinsics estimation models then found most of them struggling to achieve robust intrinsics in dynamic videos.

**Dataset Code Accessibility:**

Partly

**Dataset Code Comments:**

the dataset list can be seen from the given link. but fully accessing of the dataset currently requires log-in with institution and real-name, which may break the reviewing guidlines, so i have not tried yet.

**Ethical Considerations:**

No, there are no or only very minor ethics concerns

**Final Justification:**

After carefully reading author's rebuttal and other reviews, i think all concerns regarding this work are addressed. Thus, i raise rating as SA.

**Limitations Weaknesses:**

1. small typos exists in current version, e.g, line 107, converge

2. potential ground truth Inaccuracy, though calibrations including LUT and Kalibr are  applied. Ground truthes still relies on interpolation between sparsely calibrated points, which may not fully capture the complex, non-linear behavior of the lenses.

3. limited diversity of lens models: With all videos captured using only two lens models, the benchmark's variety is restricted, and it may not represent the diverse camera lenses, e.g, more common mobile phones.

**Strengths Contributions:**

1. Tracking per-frame camera intrinsics is important for 3D reconstruction and various 3D understanding tasks. Prior works always assume constant intrinsics over videos due to limited datasets in community.  ChangeIn annotate intrinsics per-frame for all videos and extend Kalibr to significantly improve annotation quality. This work paves the way to develop a strong camera intrinsics estimation model, is an important step to solve this problem, is a solid contribution for the community.

2. videos collected are represntative in real-world scenes and have high scene diversity, including 126 indoor scenes and 260 outdoors locations. Besides, natural cinematic shots are also recorded and annotated per-frame.

3. Common baselines including COLMAP, DroidCalib and GeoCalib is evaluated on this benchmark, showing that current benchmark struggles to accurately estimate intrinsics in videos.

4. the overall writing and presentation is in good quality, i appreciate the figures 5 and figures 6 that showing the intrinsics calibration.

---

> ### Author Rebuttal · Authors · 2025-07-31
>
> Thank you for the constructive review. Responses to key points:
>
> ---
> ### Concern 1:
> _Ground truth accuracy may be inaccurate due to LUT interpolation in regions that are sparsely calibrated or exhibit non-linear behavior._
>
> #### Response:
> We have conducted new **leave-one-out cross-validation experiments** which show that the LUT interpolation is accurate and produces reliable ground truth.
>
> In our new experiments, we withhold one calibration datapoint from our LUT at a time and evaluate the accuracy of the camera intrinsic estimate for the held out point based on interpolating among its neighboring datapoints. Across both lenses, the **median camera focal length error is 1%** (max error < 4.1%), and the **median principal point error is 0.3%** (max error < 2.5%). We will include full experimental details and results in our appendix.
>
> ---
> ### Concern 2:
> _The diversity of the benchmark is limited because all recording was done with two lens models._
>
> #### Response:
> While our benchmark uses a single camera body and two zoom lenses, this **does not limit its value for evaluating intrinsics prediction models**.
>
> Our benchmark serves as a **strong discriminator for robust dynamic intrinsics prediction models**. Robust models, which may benefit from diverse training data, should naturally perform well on our benchmark. However, less robust models may fail on our benchmark unless they are trained on data that is similar to our test set in terms of scene and hardware diversity. To make our benchmark effective for evaluation, we ensure that **no video sequence is split across validation and test sets**—each belongs entirely to one or the other.
>
> As an aside, our calibration approach also **generalizes to any lens and sensor combination that provides lens metadata, including some smartphone lenses**, so it can be used to broaden hardware diversity in future extensions. Many cinema lenses other than the ones we use also support the required metadata through standardized protocols such as **Cooke /i Technology** and **ARRI LDS**. Some **non-cinema zoom lenses** are also compatible with our calibration pipeline, although they may require additional setup through the use of **external lens motors** to track the analog state of the rings and convert it into zoom and focus values. **Mobile device lenses** typically exhibit changes in intrinsics only due to varying focus, as they rely on **digital zoom** (i.e. cropping) rather than true continuous optical zoom. Nevertheless, it is possible to construct a **1D LUT** for such lenses by using APIs such as **ARKit** on iPhone and the **Camera2 API** on Android, which provide access to focus distance information.
>
> ---
> ### Concern 3:
> _There are some typos in the manuscript._
>
> #### Response:
> Thank you for pointing this out. We will carefully review the manuscript and correct all typos in the camera-ready version.
>
> ---
>
>
>
> Thank you again for your time and feedback.

---

> > ### Comment · Reviewer_CvcE · 2025-08-07
> >
> > i appreciate authors' additional experiments and detailed rebuttal, most of my concerns have been addressed. thus, i decided to keep my positive rating.

---

### Official Review · Reviewer_VEg7 · 2025-07-02

**Rating:** 4
**Confidence:** 4

**Summary:**

The work introduces ChangeIn, a benchmark dataset with per frame ground truth camera instrinsics for real world videos. The dataset provides 386 diverse high resolution videos with 143k+ annotated frames in indoor and outdoor environments while maintaining diversity with intrinsics and camera changes. The authors evaluate baseline methods on this new presented benchmark showcasing the challenges with on dynamic instrinsic predictions.

**Dataset Code Accessibility:**

Yes

**Ethical Considerations:**

No, there are no or only very minor ethics concerns

**Final Justification:**

Raising score from 3->4 based on author rebuttal.

**Limitations Weaknesses:**

- The biggest weakness is lack of discussion regarding how this dataset can be adopted for dense vision tasks. The scarcity of datasets with camera instrinsics is well known in the community, this dataset presents an encouraging foray into addressing the issue. ChangeIn dataset can be clearly adapted for 2d and 3d dense vision tasks, and this needs to be addressed atleast in future work section. Even better would be some empirical runs on few dense vision tasks in the appendix/main draft.
- Whether this dataset can be adopted for popular vision tasks directly, or further work is needed to convert the annotations into usable form, isn't clear due to lack of discussion.Current scope of the work is extremely limited to instrinsic prediction tasks, and a vision conference is a better avenue for this paper than a machine learning conference.
- Current empirical results in section 6 is limited.

**Strengths Contributions:**

- The dataset addresses the gap in availability of 3d video datasets with camera instrinsics annotations.
- Diversity of the dataset with respect to indoor and outdoor scenes, range of camera motion and intrinsic variations.
- Creation of modified Kalibr and look up tables (LUT) using lens intrinsics.
- Clear and well written explanation of methodology and hardware, which helps reproducibility
- Baseline evaluation showcasing current methods struggling dynamic instrinsics.

---

> ### Author Rebuttal · Authors · 2025-07-31
>
> Thank you for the constructive review. Responses to key points:
>
> ---
> ### Concern 1:
> _There is no discussion of how the benchmark can be used to train or otherwise be adapted for dense 2D/3D vision tasks._
>
> #### Response:
> Our benchmark contains intrinsics annotations that can **directly be used to train downstream dense 2D/3D vision algorithms**.
>
> Many dense 2D/3D vision algorithms **predict camera intrinsics as an intermediate step**, as having accurate intrinsics often helps improve accuracy of the final output. Each image in our benchmark's validation set contains ground truth intrinsics annotations, so it can **directly be used to supervise the intrinsics prediction module** in these dense vision algorithms.
>
> ---
> ### Concern 2:
> _The current scope of the work is extremely limited because it focuses on intrinsics prediction tasks._
>
> #### Response:
> The scope of our work is not limited because it addresses a **fundamental and underexplored problem critical for many downstream 3D and dense vision tasks**.
>
> Predicting camera intrinsics is **essential for 3D understanding from images**, as intrinsics define how 3D geometry is projected onto 2D images. Yet, no existing real-world benchmark provides per-frame ground truth intrinsics for videos with dynamic intrinsics, which are commonly found in the wild. As a result, **current methods are neither trained for nor evaluated in such settings**, leading to poor performance and limiting the robustness of downstream 3D algorithms. Our benchmark is the first step to address this gap, **enabling the development and evaluation of methods that are robust to changing intrinsics**—a key step toward more reliable 3D perception in real-world conditions.
>
> ---
> ### Concern 3:
> _The paper is better suited for a vision conference rather than a machine learning conference._
>
> #### Response:
> Our work is well suited for an ML conference.
>
> According to NeurIPS policy, the NeurIPS Datasets and Benchmark track "serves as a venue for high-quality publications on **highly valuable machine learning datasets and benchmarks crucial for the development and continuous improvement of machine learning methods**." Our work presents a novel benchmark addressing a core vision problem for which many state-of-the-art methods employ machine learning, such as GeoCalib and DroidCalib (Lines 118-123). As a result, it is well suited for an ML conference.
>
> ---
> ### Concern 4:
> _There are limited empirical results._
>
> #### Response:
> Our current empirical results are not limited, but we will nevertheless continue to add more results.
>
> Our current experiments include evaluation of **state-of-the-art baselines** for intrinsics prediction, as well as **empirical results demonstrating the strong performance of our modified Kalibr calibration algorithm**. We plan to further broaden coverage by adding **additional baseline evaluations** and empirical results from **running existing dense vision methods** on our benchmark.
>
> ---
>
>
>
> Thank you again for your time and feedback.

---

### Official Review · Reviewer_Bzix · 2025-07-03

**Rating:** 5
**Confidence:** 4

**Summary:**

This paper introduces ChangeIn, a benchmark dataset for evaluating camera intrinsic estimation methods on videos with dynamic (changing) camera intrinsics. The work addresses a significant gap in existing benchmarks, which typically assume static camera intrinsics throughout video sequences. Primary contributions include:

- **Diverse benchmark with dynamic intrinsics**: it provides per-frame ground truth camera intrinsics for 143K+ frames across 386 high-resolution videos with changing intrinsics. The dataset captures diverse indoor (126 videos) and outdoor (260 videos) environments with various camera motions and intrinsic changes, including controlled zoom/focus adjustments and realistic cinematic shots.

- **Principled data collection setup**: The authors use professional cameras with zoom lenses that record lens metadata (Lens Focal Length and Focus Distance). They create per-lens lookup tables (LUTs) that map this metadata to accurate camera intrinsics through extensive calibration experiments.

- **Enhanced calibration toolbox**: They extend the Kalibr calibration toolbox with modifications to improve accuracy and robustness, addressing issues like focal length initialization and principal point drift. The pipeline combines board-based and drone-based calibration to handle different field-of-view cases.

- **Baseline evaluation**: The authors evaluate existing intrinsic estimation methods (COLMAP, DroidCalib, GeoCalib) on their benchmark, showing that current approaches struggle significantly with dynamic intrinsics prediction and more work needs to be done.

Overall, this LUT-based approach enables accurate per-frame intrinsic annotation without disrupting natural video recording, tackling the challenge of obtaining ground truth for dynamic intrinsics while maintaining video continuity and realism.

**Dataset Code Accessibility:**

Yes

**Ethical Considerations:**

No, there are no or only very minor ethics concerns

**Final Justification:**

I have read through the authors' rebuttal and also other reviewers' comments. In my opinion, this paper provides a solid contribution for an important but under-explored task of changing intrinsics and it could potentially help future works in this area. As a result, I keep my Accept rating.

**Limitations Weaknesses:**

- **Limitations of the interpolation scheme**: While Section 4.4 describes the LUT interpolation approach, the paper does not validate interpolation accuracy. The authors state they "intentionally avoid more sophisticated interpolation schemes or physics-based priors" but there is no quantitative analysis of how interpolation errors propagate to final intrinsic estimates. This could be a concern for regions with sparse calibration data/extreme cases, where interpolation quality directly affects ground truth reliability.

- **Scalability and generalization limitations**: While the hardware setup is technically impressive, it raises practical concerns around reproducibility and scalability. The reliance on specialized equipment, such as specialized cameras, RTK-enabled drone, and multiple calibration targets, creates difficulties for replication/scalability. The benchmark is also limited to a single camera body and two specific zoom lenses. Different manufacturers, sensor formats, or lens designs may have different intrinsic behaviors, so it’s unclear how well the approach generalizes beyond this specific setup. Such a sophisticated setup also limits the data diversity.

- **Potential to help learn better intrinsics prediction models**: While the dataset is clearly valuable as a benchmark, it would be helpful to understand whether it could also support training better intrinsics prediction models. Exploring this would increase the impact and utility of the dataset beyond evaluation.

**Strengths Contributions:**

- **Novel and interesting problem formulation**: The paper addresses an important and underexplored problem of dynamic camera intrinsics: most existing 3D vision algorithms assume fixed camera intrinsics across a video, which does not hold in real-world scenarios involving zoom lenses, autofocus, or other dynamic adjustments. This limitation is well-motivated with concrete examples such as DSLR cameras with zoom lenses and smartphones with active autofocus.

- **Rigorous and technically sound approach**: The proposed method for recovering per-frame intrinsics is rigorous, using specialized lens metadata (LFL and FD) and lookup tables (LUTs) to retrieve intrinsics efficiently at each frame. This avoids disrupting temporal continuity while enabling accurate modeling of dynamic intrinsics. The calibration strategy/setup is rigorous: it combines board-based calibration for small-to-medium field-of-view (FOV) settings with a drone-based setup (equipped with RTK GPS) for large-scale FOVs. This approach effectively handles the challenges posed by varying focal lengths and focus distances.

- **Dataset scale and diversity**: It captures a wide range of camera motions (e.g., static, pan, tilt, dolly) and diverse patterns of intrinsic variation (monotonic drift, periodic change, sudden shifts), significantly improving upon prior works in both realism and scale.

- **Strong technical clarity**: The paper explains calibration challenges and solutions clearly, including the concept of FOV spatial footprint. The validation of the modified Kalibr algorithm in synthetic settings adds further credibility. The supplementary material includes extensive implementation details and evaluation setups.

- **Reproducibility**: The authors provide source code and a subset of validation data.

---

> ### Author Rebuttal · Authors · 2025-07-31
>
> Thank you for the constructive review. Responses to key points:
>
> ---
> ### Concern 1:
> _LUT interpolation approach lacks quantitative analysis of its accuracy, which could affect ground truth reliability._
>
> #### Response:
> We have conducted new **leave-one-out cross-validation experiments** which show that the LUT interpolation is accurate and produces reliable ground truth.
>
> In our new experiments, we withhold one calibration datapoint from our LUT at a time and evaluate the accuracy of the camera intrinsic estimate for the held out point based on interpolating among its neighboring datapoints. Across both lenses, the **median camera focal length error is 1%** (max error < 4.1%), and the **median principal point error is 0.3%** (max error < 2.5%). We will include full experimental details and results in our appendix.
>
> ---
> ### Concern 2:
> _Specialized hardware used in calibration approach could limit the reproducibility and scalability of the benchmark._
>
> #### Response:
> The specialized hardware does not limit the reproducibility or scalability of the benchmark, as it is all **commercially available for rent or purchase** and enabled us to efficiently complete calibration of both lenses **within a few days and on a reasonable budget**.
>
> For reproducibility, all the camera equipment can be rented via global or online providers that offer shipping. The RTK, drone kit, and calibration targets are all commercially purchasable online. We list vendor details in Lines 187-194 in the main paper and in the caption of Figure 3 caption in the supplementary.
>
> Our approach is significantly more scalable than methods that require frame-by-frame recalibration of zoom lenses (Lines 126-129). **LUT construction is a one-time cost and only takes a few days per lens**. Once built, the LUT can be used with any future footage captured using the same lens. Furthermore, the procedure can be adapted to **any lens that provides lens focal length (LFL) and focus distance (FD) metadata**, which extends beyond the two lenses included in this benchmark.
>
> ---
> ### Concern 3:
> _Specialized hardware used in calibration approach raises concerns about generalizability to other camera and lens setups, as well as the hardware diversity of the benchmark._
>
> #### Response:
> Our calibration approach **generalizes to any lens and sensor combination that provides lens metadata**, including less expensive non-cinema lenses, and can be used to broaden hardware diversity in future extensions.
>
> Many cinema lenses and cameras other than the ones we used also support the required focus, iris, and zoom (FIZ) metadata through standardized protocols such as **Cooke /i Technology** and **ARRI LDS**. Manufacturers that provide compatible lenses include **Cooke, ARRI, Zeiss, Angenieux,** and **Fujinon**. These lenses are typically compatible with **PL-mount cameras** from a wide range of manufacturers, including **ARRI, RED, Panavision, Blackmagic, Sony**, and more.
>
> Some **non-cinema zoom lenses** are also compatible with our calibration pipeline, although they may require additional setup. For lenses that have mechanical zoom and focus rings but do not directly report metadata, **external lens motors** can be used to track the analog state of the rings and convert it into zoom and focus values required by our LUT-based calibration method.
>
> ---
> ### Concern 4:
>
> _Whether the dataset can be used to support training better intrinsics prediction models is not explored._
>
> #### Response:
> The validation split of our benchmark **could potentially serve as training data** for intrinsics prediction models, but it is not clear whether the results will be satisfactory without additional images or scenes.
>
> If additional data is needed, our calibration method and modified Kalibr algorithm can be used to extend the benchmark. We will add discussion of this in our future works section.
>
> ---
>
>
>
> Thank you again for your time and feedback.

---

### Decision · Program_Chairs · 2025-09-18

**Decision:**

Accept (poster)

**Comment:**

This paper captured a dataset for the task of intrinsics per-frame calibration, containing 386 diverse videos (more than 143K frames) with varying intra-video intrinsics and per-frame ground truth. The opinion of the reviewers and of the AC is generally positive. The dataset targets a task that is both relevant and underexplored (most 3D vision works are evaluated mostly with constant intravideo intrinsics), and the authors nicely motivate it by showing the modest results obtained by several recent baselines at the task. While providing per-frame ground truth is challenging, the authors address that in an elegant, principled and sound manner, both technically and methodologically.

The authors addressed satisfactorily reasonable concerns from the reviewers, namely reproducibility due to the use of specialized hardware, potential biases due to the use of just two camera lenses or missing references. Specifically, the authors are urged to add to the paper cross-validation results for the LUT, to support the quality of the ground truth.